# Defects in nephrogenesis result in an expansion of the *Foxd1+* stromal progenitor population

Michael G. Michalopulos[1,*], Yan Liu[2], Dinesh Ravindra Raju[2], John T. Lafin[3], Yanru Ma[4], Dhruv Gaur[1,‡], Sadiksha Khadka[1], Chao Xing[5], Andrew P. McMahon[6,§], Thomas J. Carroll[7] and Keri A. Drake[1,¶]

## ABSTRACT

The *Foxd1+* stromal progenitor cells give rise to the majority of the renal interstitium; yet, much remains to be understood about how this self-renewing progenitor population is regulated during development. Here, we demonstrate that disruption of the nephron progenitor cell (NPC) lineage via loss of *Wt1* (i.e. *Six2cre;Wt1c/c*) results in an expansion of *Foxd1+* progenitor cells in mice. Analyses of two additional models (i.e. *Wnt4-null* mutants, which fail to form nephron structures similar to *Six2cre;Wt1c/c* kidneys, and NPC ablation via diphtheria toxin using the *Six2cre;RosaDTAc/+*) phenocopy the expansion in *Foxd1+* cells and further confirm that mutant kidneys with defects in nephrogenesis develop an abnormal increase in the stromal progenitor population. Furthermore, single nuclei RNA-sequencing shows transcriptional changes in the *Foxd1+* progenitor cells from *Six2cre;Wt1c/c* kidneys and identifies a distinct subcluster of the *Foxd1+* stroma, which is maintained independent of signals from the nephrogenic niche in the *Six2cre; RosaDTAc/+* model. Overall, these findings provide insights into the developmental regulation of the stromal progenitor population and uncover heterogeneity within the *Foxd1+* cells, which undergo both cellular and molecular changes in response to defects in nephrogenesis.

KEY WORDS: Stromal progenitor cells, Foxd1, Wt1, Wnt4, Kidney development, Mouse

[1]Division of Pediatric Nephrology, University of Texas Southwestern Medical Center, Dallas, TX 75390, USA. [2]Eugene McDermott Center for Human Growth and Development, University of Texas Southwestern Medical Center, Dallas, TX 75390, USA. [3]Department of Urology, University of Texas Southwestern Medical Center, Dallas, TX 75390, USA. [4]Lyda Hill Department of Bioinformatics, University of Texas Southwestern Medical Center, Dallas, TX 75390, USA. [5]Department of Bioinformatics and O'Donnell School of Public Health, University of Texas Southwestern Medical Center, Dallas, TX 75390, USA. [6]Department of Stem Cell Biology and Regenerative Medicine, Eli and Edythe Broad Center for Regenerative Medicine and Stem Cell Research, Keck School of Medicine of the University of Southern California, Los Angeles, CA 90033, USA. [7]Department of Internal Medicine, Division of Nephrology, University of Texas Southwestern Medical Center, Dallas, TX 75390, USA.
*Present address: Department of Pediatrics, Division of Pediatric Nephrology, University of Iowa Carver College of Medicine, Iowa City, IA 52242, USA. ‡Present address: Midwestern University, Chicago College of Osteopathic Medicine, Chicago, IL 60515, USA. §Present address: Division of Biology and Biological Engineering, California Institute of Technology, Pasadena, CA 91125, USA.

¶Author for correspondence (Keri.Drake@UTSouthwestern.edu)

 K.A.D., 0000-0002-9355-5202

## INTRODUCTION

The development of the mammalian kidney is a highly coordinated process involving reciprocal signaling interactions among the nephron, ureteric and stromal progenitor lineages. Such cell-lineage crosstalk has been shown to regulate multiple aspects of kidney development, from directing initial events in the formation of the metanephros via invasion of the ureteric bud (UB) into the metanephric mesenchyme (Grobstein, 1953, 1956), to regulating the self-renewal, proliferation and differentiation of nephron progenitor cells (NPCs) and UB branching essential in determining final nephron allotment (Barak et al., 2012; Carroll et al., 2005; Cebrian et al., 2014; Costantini and Kopan, 2010). However, the extent to which cell-autonomous versus non-autonomous mechanisms regulate the complex processes in normal kidney development is still being understood.

The self-renewing *Foxd1+* stromal progenitor population gives rise to the majority of the renal interstitium, including pericytes, mesangial cells and vascular smooth muscle (Kobayashi et al., 2014). Signals from the embryonic stroma have been shown to regulate NPC maintenance/differentiation, UB branching and patterning of the vasculature (Das et al., 2013; Hatini et al., 1996; Hum et al., 2014; Levinson et al., 2005). Additionally, the integration of stromal cells in kidney organoids has further confirmed the importance of the stroma in generating 'higher-order structures' and recapitulating *in-vivo* development (Tanigawa et al., 2022). Despite this exciting progress, much remains to be understood about mechanisms regulating stromal progenitor maintenance and differentiation, as well as potential contributions to disease states that may result from stromal defects in both developing and adult kidneys.

While numerous studies have demonstrated that signals from the stroma regulate NPC differentiation (D'Cruz et al., 2023; Das et al., 2013; Drake et al., 2020; Rowan et al., 2018), much less is known about how signals from the NPC lineage may influence stromal development. Previous single cell RNA-sequencing studies have bioinformatically identified potential receptor-ligand interactions (Combes et al., 2019), suggesting that NPCs may signal to the stroma during normal development. However, here we sought to evaluate stromal development in mouse models with targeted mutations to the NPCs to investigate how defects in the nephron lineage may non-autonomously affect maintenance and differentiation of the stromal progenitor cells. To do this, three genetically engineered mouse models were examined, including: (1) *Six2cre;Wt1c/c* mutants, with ablation of *Wt1* function in the NPCs known to block NPC differentiation while maintaining the self-renewing NPC population (Berry et al., 2015; Davies et al., 2004; Essafi et al., 2011; Hartwig et al., 2010); (2) *Wnt4-null* mutants, which have also been shown to lack NPC differentiation due to a block in mesenchyme-to-epithelial (MET) transition (Kispert et al., 1998); and (3) NPC ablation via diphtheria toxin via *Six2cre;*

$RosaDTA^{c/+}$. In all three models, we observe an expansion of the nephrogenic zone stroma confirmed by immunofluorescence (IF) and *in-situ* hybridization (ISH), demonstrating that defects in nephrogenesis result in an accumulation of the $Foxd1^+$ stromal progenitor population. Single nuclei RNA-sequencing (snRNA-seq) from embryonic day (E)15.5 control and $Six2cre;Wt1^{c/c}$ mutant kidneys identifies heterogeneity of $Foxd1^+$ stroma with differentially expressed genes (DEGs) in distinct subclusters, suggesting that non-autonomous effects from defects in nephrogenesis result in both cellular and molecular changes in the stromal progenitor population. Furthermore, human fetal kidney snRNA-seq suggests that distinct subclusters of the murine stromal progenitor cells are conserved in human fetal kidneys as well. Overall, the findings from this study demonstrate heterogeneity of the $Foxd1^+$ progenitor cells and identifies an abnormal cellular and molecular response when nephrogenesis is disrupted, thus providing additional insights into mechanisms regulating the stromal progenitor population and potential implications for advancing the understanding of how stromal cells contribute to both normal development and disease.

## RESULTS

### Genetically engineered mouse models with defects in nephrogenesis show an expansion of the nephrogenic zone stroma

Previous studies have shown that loss of *Wt1* in the nephron lineage prevents nephrogenesis via failed MET through the transcriptional regulation of *Wnt4* (Berry et al., 2015; Davies et al., 2004; Essafi et al., 2011). Here, we used a conditional *Wt1* allele with *loxP* sites flanking exons 8 and 9 of the DNA binding domain (Gao et al., 2006) and *Six2cre* to ablate its function within the NPCs (Kobayashi et al., 2008). Consistent with previously published observations, $Six2cre; Wt1^{c/c}$ mutant kidneys maintained sine oculis-related homeobox 2 (SIX2)-positive NPCs (Fig. 1J,N,R; Fig. S1) and showed defects in NPC differentiation (Fig. 1N, with neural cell adhesion molecule, NCAM, expression labeling early differentiating structures marked by arrows, and Fig. S1F showing loss of LIM homeobox 1, LHX1, structures marked by arrowheads). This resulted in smaller kidney size compared to controls (Fig. 1F versus 1E, respectively; quantified in Fig. S1M). Despite these observed defects in nephrogenesis, some differentiated structures including proximal tubules and glomeruli still formed, most notable at later timepoints (Fig. S1B, arrows), and presumably due to incomplete efficiency of the *Six2cre* model with early NPCs 'escaping' cre-recombination, as previously reported (Marable et al., 2018). Upon further examination of $Wt1^{c/c}$ mutant kidneys, we observed an expansion of cells in the nephrogenic zone surrounding the cap mesenchyme on Hematoxylin and Eosin (H&E) staining compared to controls (Fig. 1F versus 1E, respectively). IF co-staining of the pan-stromal marker meis homeobox 1, 2, 3 (MEIS1/2/3) with SIX2 labeling of the NPCs showed a 'multi-layering' of stromal cells between the outer cap mesenchyme and periphery of the kidney compared to controls (Fig. 1J versus 1I, respectively; arrowhead). Additionally, IF of NCAM staining in $Six2cre;Wt1^{c/c}$ mutant kidneys showed abnormal expression in the nephrogenic zone stroma surrounding mutant NPCs. IF of aldehyde dehydrogenase family 1, subfamily A2 (ALDH1A2), which is specifically expressed by the nephrogenic zone stroma surrounding the NPCs/UB, also appeared to be expanded compared to controls (Fig. 1N,R versus 1M,Q, arrowheads, respectively). Quantification of $ALDH1A2^+$ stroma at the outer periphery of the kidney confirmed a statistically significant expansion compared to cre-negative controls (Fig. 1V). Inclusion of a $Rosa26^{EYFP}$ reporter was used to confirm that *Six2cre* specifically targets the NPC lineage, with no

recombination observed in the stroma (Fig. S1H), thus showing no evidence suggesting that NPCs may 'transdifferentiate' into stroma. Additionally, an obvious increase in proliferation was not observed in mutant kidneys (Fig. S1J,L). These data reveal that *Wt1* ablation targeted to the NPCs resulted in an expansion of the stroma within the nephrogenic zone where the $Foxd1^+$ progenitor cells reside, suggesting that abnormal stromal progenitor maintenance and/or differentiation may occur in response to defects in NPC differentiation.

Given this finding, we postulated that an expansion of the stromal progenitor cells may result from potential disruptions in non-autonomous signaling from the nephron-to-stromal lineages, versus the possibility that global defects in development resulting in smaller kidneys and/or an 'immature' stromal phenotype may contribute to the phenotype, and also raised the question whether this finding is specific to $Six2cre;Wt1^{c/c}$ mutants. To begin to address these possibilities, we first examined additional mutant models disrupting the nephron lineage to evaluate whether effects on the nephrogenic zone stroma are specific to $Six2cre;Wt1^{c/c}$ mutants or may be seen in other models with defects in nephrogenesis. As described in previous publications, loss-of-function of *Wt1* in the nephron lineage both prevents NPC differentiation (with a block in MET and lack of renal vesicle formation) and also affects gene expression in the self-renewing NPCs, but has not been previously reported to show abnormal stromal development as identified here. Past studies showed significant gene expression changes in *Wt1* mutant NPCs, with upregulation of genes involved in muscle and cartilage/ bone regulation (Berry et al., 2015) and downregulation of known NPC regulatory genes (Hartwig et al., 2010; Motamedi et al., 2014). Accordingly, the $Six2cre;Wt1^{c/c}$ model may perturb stromal development as a result of the loss of differentiated NPCs versus abnormal signals from *Wt1* mutant self-renewing NPCs. Thus, in order to gain insights into the cellular mechanism underlying this observed stromal expansion, we evaluated two additional mutant mouse models, including *Wnt4-null*, which targets early differentiated NPCs to block nephrogenesis, as well as $Six2cre;RosaDTA^{c/+}$, which results in NPC ablation and thus targeting of the nephron lineage at this earliest stage.

As described above, *Wnt4-null* mutant kidneys have been previously shown to have defects in NPC differentiation/ tubulogenesis at the stage of renal vesicle formation without targeting the self-renewing NPC population (Kispert et al., 1998; Stark et al., 1994). As expected, mutant kidneys showed severe defects in development, with decreased size (Fig. 1C) and a lack of differentiating structures (Fig. 1G,O). Expansion of the nephrogenic zone stroma by MEIS1/2/3, NCAM and ALDH1A2 expression at the outer periphery of the kidney also occurred in this model (Fig. 1K,O,S, respectively), similar to $Six2cre;Wt1^{c/c}$ mutants.

NPC ablation using $Six2cre;DTA^{c/+}$ transgenic mice results in the targeted loss of $SIX2^+$ NPCs via apoptosis triggered by cre-inducible DTA (diphtheria toxin fragment A) (Brockschnieder et al., 2004). As expected, mutant kidneys showed severely perturbed development compared to controls (Fig. 1D,H versus 1A,E, respectively), with few UB branches and scattered NPCs present (Fig. 1L,P,T), presumably due to incomplete efficiency of the *Six2cre* model as described above (Marable et al., 2018). Similar to $Six2cre;Wt1^{c/c}$ and *Wnt4-null* mutants, IF of both NCAM and ALDH1A2 expression appeared to be expanded in the outer stroma at the periphery of the kidney, both in regions showing residual NPCs as well as in regions with complete NPC ablation (Fig. 1P,T, respectively; arrowheads denoting regions with maintained NPCs). Thus, these findings phenocopy both the $Six2cre;Wt1^{c/c}$ and *Wnt4-null* models demonstrating stromal expansion, even in regions without NPCs/UB, arguing against an 'ectopic' signal from abnormal *Wt1* mutant NPCs driving stromal

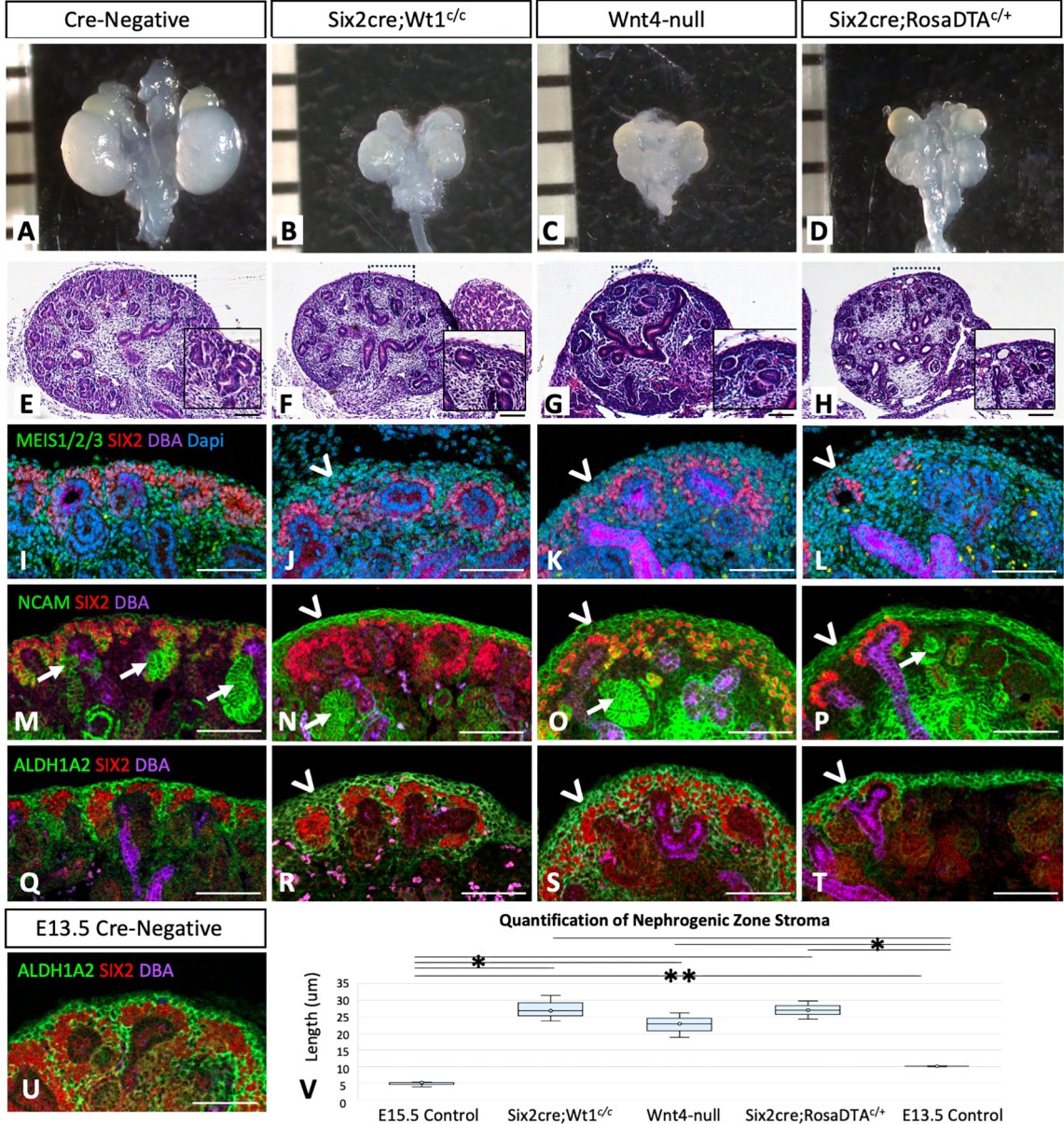

**Fig. 1. Mutant mouse models with defects in nephrogenesis show an expansion of stroma in the nephrogenic zone.** (A-D) Compared to control kidneys (A), mutant mouse models with defects in nephrogenesis including *Six2cre;Wt1c/c*, *Wnt4-null* and *Six2cre;RosaDTAc/+* (B-D, respectively) result in decreased kidney size at E15.5. (E-H) Histology confirms fewer nephron structures in mutant models. (I-U) Immunofluorescence (IF) of E15.5 control and mutant kidneys show NPCs labeled with SIX2 (red), early differentiating nephron structures labeled with NCAM (green, M-P; arrows), ureteric bud/collecting ducts labeled with *Dolichos biflorus* agglutinin (DBA; purple), as well as stroma labeled with MEIS1/2/3 as a global stromal marker (green, I-L) and ALDH1A2 specifically expressed in the stroma surrounding the cap mesenchyme, or 'nephrogenic zone' (green, Q-U). Mutant kidneys with defects in nephrogenesis show expansion of stromal cells at the outer periphery of the kidney (arrowheads), with abnormal expression of NCAM (M-P) and expression ALDH1A2, a marker of the nephrogenic zone stroma (Q-T). ALDH1A2 expression at E13.5 (U) does not show similar expansion to mutant kidneys. (V) Quantification of the nephrogenic zone stroma on sections of control and mutant kidneys stained with ALDH1A2 and SIX2 (i.e. as shown in panels Q-T) shows a statistically significant difference between E13.5 and E15.5 control kidneys (**P=0.03), as well as control and mutant kidneys (*P<0.001), with no statistically significant differences noted among the three mutant models. Box plots show median line and first to third interquartile ranges; whiskers indicate 1.5× the interquartile ranges. Representative images are shown from n=3 embryos. Scale bars: 100 µm (E-H); 50 µm (I-U).

expansion. This was confirmed via quantification of the ALDH1A2+ stroma in control and mutant kidneys, with no differences seen among the three mutant models (Fig. 1V).

Notably, all three mutant models show grossly smaller kidneys with a lack of differentiating structures and fewer UB branches, consistent with global defects in development. This raises the possibility that the stromal phenotype in mutant kidneys may result from a number of factors, including: (1) global developmental delay, potentially exhibiting an 'immature' stromal phenotype which may be normally present at earlier timepoints; (2) continued self-renewal of the stromal progenitor population despite a lack of normal kidney size/growth; and (3) a possible defect in normal stromal differentiation – all of which may result in an 'abnormal accumulation' of the self-renewing stromal progenitor population. In an effort to evaluate these possibilities, we first compared ALDH1A2 expression in the nephrogenic stroma of wild-type

kidneys at E13.5 to control and mutant kidneys at E15.5, with the rationale that if the same stromal phenotype was observed at earlier time points in development, then a delay in development/reduced branching may contribute to the stromal phenotype in mutant kidneys. Quantification of ALDH1A2 expression in wild-type E13.5 kidneys (Fig. 1U) did show a greater width than wild-type E15.5 kidneys (Fig. 1V, *P*-value=0.03). However, the width of ALDH1A2 expression in all three mutant models at E15.5 demonstrated a significantly greater increase in comparison to both E15.5 and E13.5 control kidneys (*P*-value<0.001), with the average measurement from E13.5 wild-type kidneys being 10.1 µm, which was less than half of the expanded stromal measurements in the three mutant models, which averaged 27.4, 22.6 and 27.1 µm in the *Six2cre;Wt1$^{c/c}$*, *Wnt4-null* and *Six2cre;RosaDTA$^{c/+}$* mutants, respectively. These findings suggest that the abnormal expansion of the nephrogenic stroma in mutant kidneys is unlikely to be solely due to a secondary defect reflective of an 'immature' stromal phenotype normally found at earlier time points in development. However, the expansion of the nephrogenic stroma could potentially result from self-renewing progenitor cells that 'accumulate' over time due to either an inability to differentiate and/or a lack in overall kidney growth due to defects in nephrogenesis, which we attempt to evaluate further as described below.

### Single nuclei RNA-seq identifies subclusters of the *Foxd1$^+$* stromal progenitor population in control and *Six2cre;Wt1$^{c/c}$* mutant kidneys

The study of stromal progenitor cell maintenance/differentiation has been limited by a lack of validated markers for these cell types. While past reports identified cellular heterogeneity within the *Foxd1$^+$* stromal progenitor population at E18.5 (England et al., 2020), we performed snRNA-seq at E15.5 to attempt to identify additional markers of the *Foxd1$^+$* cells at this time point and to evaluate for transcriptional differences in the stroma of *Six2cre; Wt1$^{c/c}$* mutant kidneys versus controls. Mice carrying the *Six2cre* transgene were used as controls for this experiment given the reported effects on nephron development in this model (Perl et al., 2024). A total of 11,906 nuclei from control kidneys and 13,291 nuclei from mutant kidneys were analyzed after quality control filtering (Fig. 2A,B). Unsupervised cell clustering identified a total of 25 clusters, with 20 corresponding to specific cell types (Fig. 2C; Table S1) based on marker gene analysis and anchor gene expression (Chaney et al., 2022). Five clusters (i.e. clusters 17, 18, 22, 23 and 24) showed either a high doublet score and/or low numbers of nuclei captured (as shown in Fig. S2) and thus were not included in subsequent analyses.

Focusing on the stroma, three clusters (i.e. clusters 14, 1 and 7) showed enriched expression of *Foxd1*, *Ntn1* and *Aldh1a2*, consistent with the stromal progenitor population. Additional markers enriched in these subclusters were identified and appeared to be similarly expressed in control and mutants (Fig. 2D,E, respectively). We next evaluated DEGs among the three *Foxd1$^+$* subclusters, with enriched expression of *Smoc2*, *Gria1*, *Cntnap2*, *Postn*, *Fap*, *Rxfp2*, *Rspo3*, *Grm8*, *Dlk1*, *Col6a6*, *Dkk2*, *Igf1*, *Ptger3* and *Prrx1* in cluster 14, versus *Ptn*, *Reln*, *Lsamp*, *Gdnf* and *Mcf2* in clusters 1 and 7, and *Top2a*, *Mki67* and *Cenpp* in cluster 7 (Fig. 2D,E; Table S2). These findings demonstrate a distinct transcriptional profile in cluster 14 versus cluster 1, with cluster 7 showing enrichment of cell cycle genes consistent with a proliferative cell type. Additionally, five other stromal clusters were identified (i.e. clusters 19, 2, 11, 5 and 10) showing gene expression profiles consistent with stroma from the cortical and medullary

regions of the kidney. Specifically, expression of cortical markers *Gucy1a1*, *Clca3a1* and *Ace2* are enriched in clusters 2 and 11, and medullary markers *Lrriq1*, *Col23a1*, *Alx1* and *Apcdd1* are enriched in clusters 5 and 10 (Combes et al., 2019; England et al., 2020). Thus, our snRNA-seq analysis identifies a total of eight clusters of stroma, demonstrates heterogeneity of the *Foxd1$^+$* progenitor population, and provides additional markers to further evaluate in our mutant kidney models.

Given the stromal expansion identified *Six2cre;Wt1$^{c/c}$* kidneys shown in Fig. 1, we also sought to evaluate the relative proportions of cell types captured via snRNA-seq in control versus mutant kidneys (Table S3; Fig. S3), given that our robust dataset included three separate sequencing runs using pooled kidneys from one control and one mutant embryo in each run performed. We found similar proportions of NPCs and ureteric epithelium captured in control and mutant kidneys. However, mutant kidneys showed decreased podocytes, proximal tubules and loop of Henle precursors with an increased proportion of stromal progenitor nuclei compared to the total cell types captured. Specifically, control versus mutants showed: 1.6% versus 6.0% for cluster 14, 5.0% versus 12.1% for cluster 1, and 4.7% versus 6.6% for cluster 7. These findings are consistent with the expansion of these cells demonstrated by IF and interestingly show that the percentage of proliferating cells (i.e. cluster 7) within the total *Foxd1$^+$* population was lower in mutants compared with controls (i.e. 26% versus 41%).

### The *Foxd1$^+$* stromal progenitor population, including subcluster 14, is abnormally expanded in mutant kidneys with defects in nephrogenesis

We next used ISH to examine the expression of gene markers of the *Foxd1$^+$* stromal progenitor population identified from the snRNA-seq analysis in the three mutant models compared to cre-negative and/or wild-type littermate controls (Fig. 3). Indeed, genes expressed in all three subclusters of the stromal progenitor population, including *Foxd1*, *Ntn1*, *Ebf3* and *Aox3*, showed expanded/increased expression in all three mutant models. Additionally, mutant kidneys show more prominent expression of specific markers of subcluster 14, including *Fap*, *Igf1* and *Gria1* at the very outer periphery of mutant kidneys, with this same region showing a lack of expression of *Gdnf*, which is consistent with the findings from our snRNA-seq analysis showing upregulation of *Gdnf* in subclusters 1 and 7 and minimal expression in subcluster 14 (Fig. 2D,E). Overall, this ISH data validates the expression pattern of several genes specific to the *Foxd1$^+$* population and further confirms an expansion of stromal progenitor cells in mutant kidneys with defects in nephrogenesis.

Given the possibility that the stromal patterning of mutant kidneys could reflect a 'global developmental delay' as discussed above, we sought to further evaluate the expression of our additionally identified markers of the stromal progenitor population at earlier time points in development (i.e. E13.5). Consistent with IF findings from ALDH1A2 expression and quantification shown in Fig. 1, ISH of *Foxd1*, *Ntn1* and *Aox3* showed restricted expression in control E13.5 kidneys in comparison to mutant kidneys at E15.5 (Fig. 3), further demonstrating that the expansion of the stromal progenitor population in mutant kidneys is not solely due to 'immature' stromal patterning. However, *Ebf3*, *Fap* and *Gria1* did appear to be somewhat expanded in control kidneys at E13.5, in comparison to more restricted expression in control kidneys at E15.5. Furthermore, *Gdnf* showed lower levels of expression and appeared to be excluded from the outer peripheral cells of the nephrogenic zone stroma of E13.5 control kidneys (Fig. 3N'). Given this, we systematically

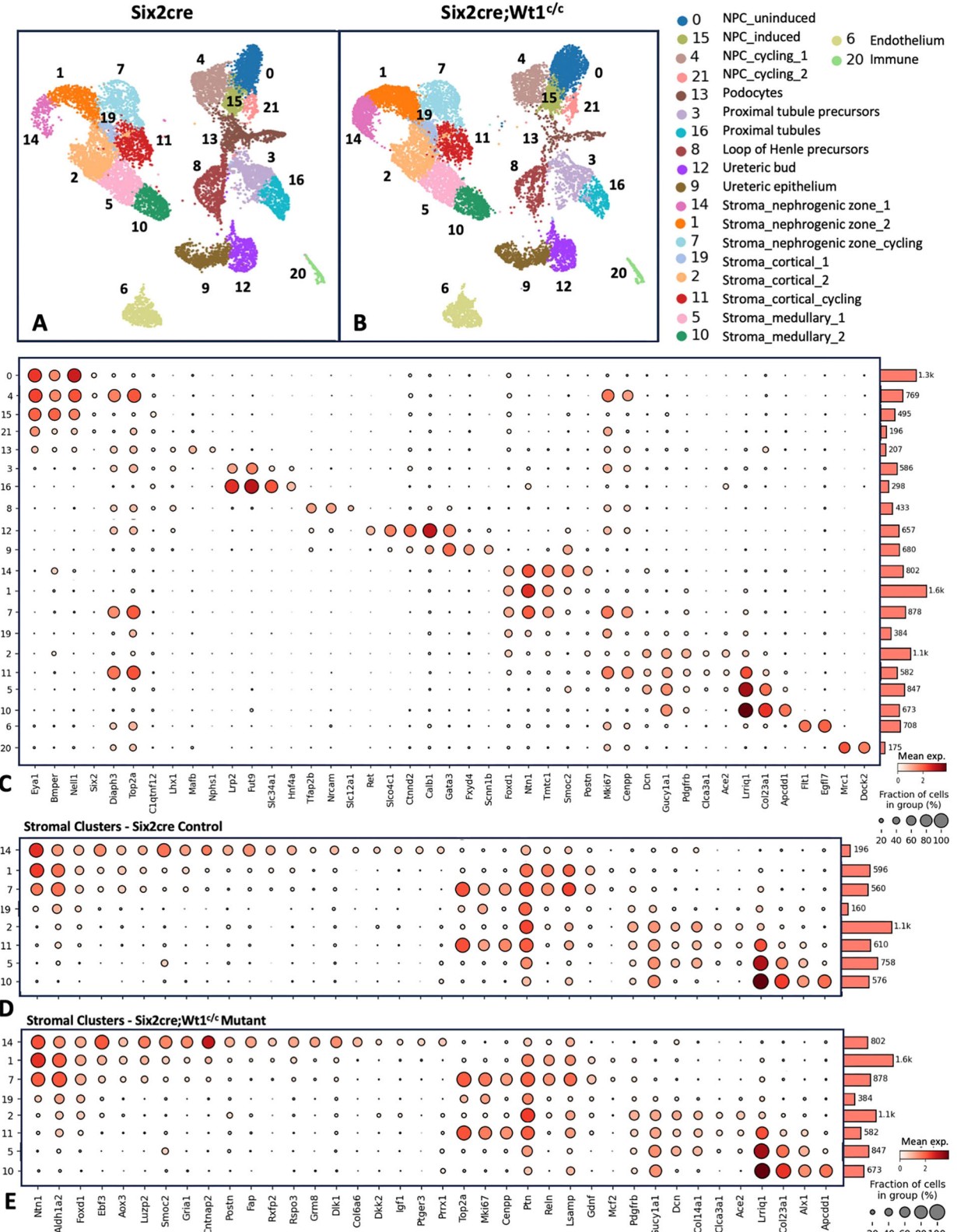

**Fig. 2. Single nuclei RNA-seq of E15.5 control (*Six2cre*) and mutant (*Six2cre;Wt1*<sup>c/c</sup>) kidneys identifies heterogeneity of the *Foxd1*+ stromal progenitor population.** (A,B) A total of 11,906 nuclei from control kidneys (A) and 13,291 nuclei from *Six2cre;Wt1*<sup>c/c</sup> mutant kidneys (B) were analyzed via unsupervised cell clustering (*n*=3 separate embryos of paired kidneys for each genotype). (C) Cell type-specific markers from control kidneys were used to identify specific cell types of 20 clusters with representative genes shown via dot plot. (D,E) Stromal clusters were specifically evaluated for gene expression profiles distinguishing the eight clusters identified. Enriched *Foxd1*+ expression was identified in three subclusters (i.e. 14, 1 and 7) consistent with the stromal progenitor population. Transcriptional profiles shown via dot plot distinguish subclusters 14 and 1, along with enriched proliferative/cell cycle markers in subcluster 7, demonstrating heterogeneity within the *Foxd1*+ stromal progenitor population of control kidneys (D), with these gene markers showing a similar expression pattern in the stromal progenitor cells of mutant kidneys (E).

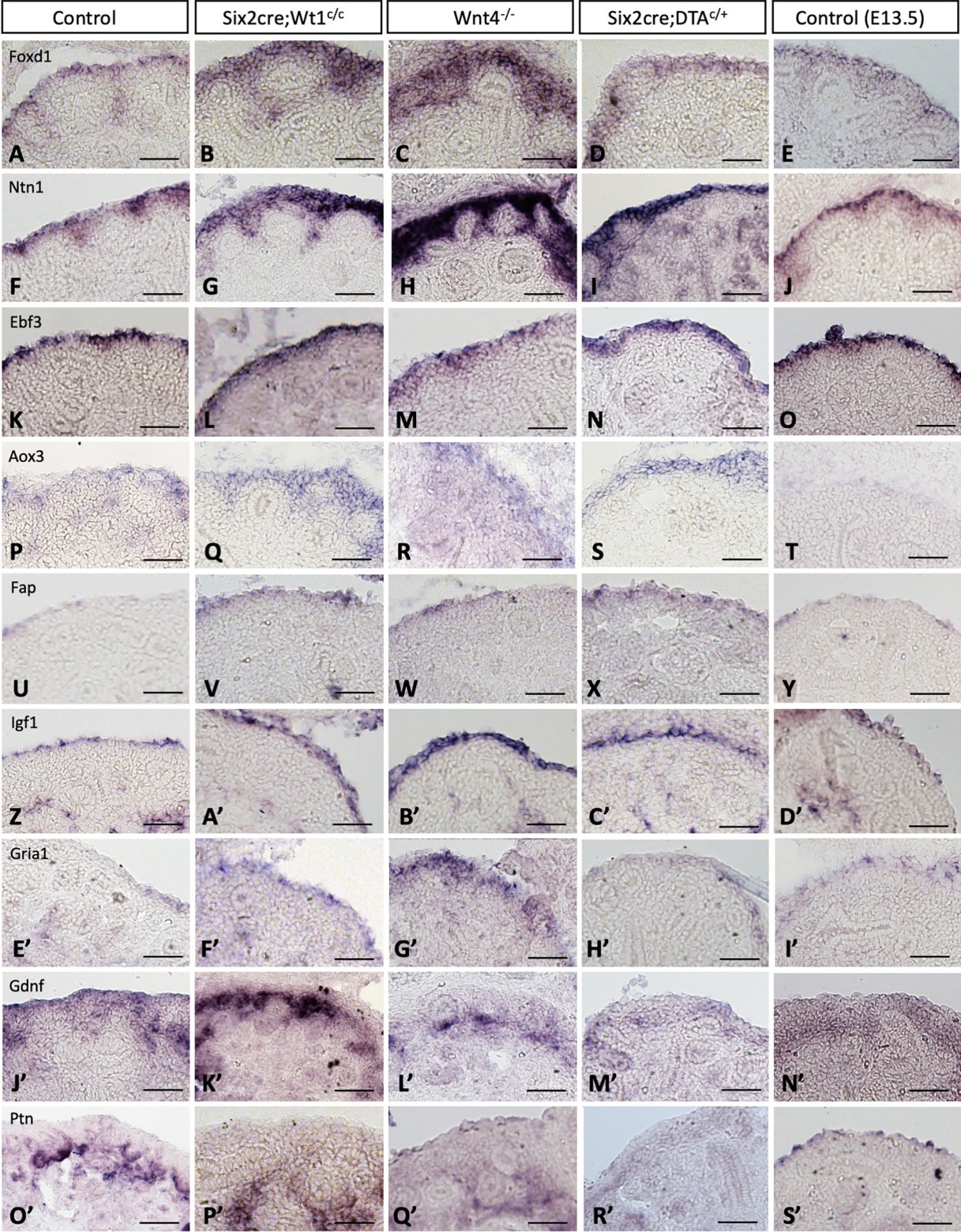

**Fig. 3. The *Foxd1*⁺ stromal progenitor population shows abnormal patterning in mutant kidneys with defects in nephrogenesis.** *In-situ* hybridization (ISH) of known markers of the nephrogenic zone stroma, *Foxd1* and *Ntn1*, as well as markers identified from snRNA-seq analyses including *Ebf3*, *Aox3*, *Fap*, *Igf1* and *Gria1*, all show expanded/increased expression in *Six2cre;Wt1^{c/c}*, *Wnt4-null* and *Six2cre;RosaDTA^{c/+}* mutant compared to control kidneys at E15.5. Localization of *Gdnf* expression shows minimal signal in the outer periphery of the nephrogenic stroma of mutant kidneys versus controls, with a similar expression pattern of *Ptn*, suggesting an expansion of subcluster 14 of the *Foxd1*⁺ stroma identified from our snRNA-seq dataset showing *Fap*⁺, *Igf1*⁺, *Gria1*⁺ and *Gdnf*⁻. When comparing the expression of these markers to E13.5 control kidneys, *Foxd1* and *Ntn1* appear to be abnormally expanded in mutant kidneys versus E13.5 controls. However, *Fap*, *Igf1*, *Gria1* and *Gdnf* appear similar to mutant kidneys, suggesting that subcluster 14 may be expanded at earlier developmental time points, yet the restricted expression of *Foxd1* and *Ntn1* at E13.5 indicates abnormal patterning of stromal progenitor cells in E15.5 mutant kidneys. Representative images are shown from *n*=3 embryos. Scale bars: 50 μm.

evaluated ISH expression of *Foxd1*, *Ntn1*, *Ebf3*, *Aox3*, *Fap*, *Igf1*, *Gria1*, *Gdnf* and *Ptn* throughout development at E12.5, E13.5, E15.5 and E18.5 (Fig. S4) with *Ntn1*, *Ebf3* and *Aox3* showing more restricted expression in late versus early development. While these findings raise the possibility that the expansion of subcluster 14 in mutant kidneys may reflect some degree of an 'immature' phenotype of the stromal progenitor population, the abnormal expansion of other markers including *Foxd1*, *Ntn1* and *Aox3* in mutant kidneys (which are not seen at earlier time points E12.5 and E13.5) suggests that abnormal changes/patterning occur in the stromal progenitor population due to defects in nephrogenesis.

### *Foxd1*⁺ stromal progenitor cells are maintained independent of signals from the nephrogenic niche

As described earlier, the *Six2cre;RosaDTA*$^{c/+}$ model results in incomplete NPC ablation, with some regions of the mutant kidney showing maintained UBs with surrounding NPCs while other regions are completely devoid of NPC/UB niches (Fig. 1L,P,T). This model thus allows us to further assess whether the expansion of stromal progenitor cells depends on signals from NPC/UB niches (i.e. areas where these structures are maintained) or whether it may occur independent of signals from the NPCs/UB. Additionally, we sought to use this model to evaluate how severe defects in nephrogenesis resulting in 'stunted/reduced' overall kidney size may influence the development of the stromal progenitor population.

Towards these goals, we first evaluated overall kidney size in control and *Six2cre;RosaDTA*$^{c/+}$ mutant kidneys at E15.5 and E18.5 (Fig. 4A-D), with no difference noted in size at E13.5 (data not shown) likely due to the slight delay in NPC ablation and/or inefficient cre in this model as described above. Quantification of kidney length (Fig. 4E) showed significantly decreased kidney size at E15.5 compared to cre-negative littermate controls, with minimal interval growth of mutant kidneys from E15.5 to E18.5. However, evaluation of the nephrogenic zone stroma in mutant kidneys interestingly showed a significant expansion in E15.5 mutant kidneys (Fig. 4G) that subsequently improved by E18.5 (Fig. 4I) compared to controls (Fig. 4F versus 4H, respectively), as quantified by measuring the outer periphery stroma by ALDH1A2 staining (Fig. 4J). Analysis of this most outer stromal layer showed no difference in areas where NPCs/UB are maintained versus regions of mutant kidneys where there was complete ablation confirmed by serial sections (data not shown). However, ALDH1A2 staining showed nephrogenic zone stromal cells surrounding the cap mesenchyme in regions where NPCs were still maintained and thus appeared to be significantly expanded (Fig. 4G,I, with arrows showing regions of maintained NPCs/UB versus arrowheads showing regions with complete NPC ablation). Furthermore, ISH showed expression of *Foxd1*, *Ntn1*, *Ebf3*, *Fap* and *Igf1* with a lack of *Gdnf* expression compared to controls (Fig. 4K-V) in the outer periphery stroma of mutant kidneys, suggesting that the *Foxd1*⁺ stromal progenitor population maintained in *Six2cre;RosaDTA*$^{c/+}$ mutant kidneys shows an expression pattern consistent with cluster 14, even in regions with complete loss of the NPCs/UB of the nephrogenic niche. Thus, the *Six2cre;RosaDTA*$^{c/+}$ model demonstrates that the maintenance of stromal progenitor cells, and specifically cluster 14, does not require signals from the NPCs/UB. Additionally, findings from this model suggest that the abnormal expansion of stromal progenitor cells at E15.5 may result from continued proliferation of the stroma that occurs regardless of reduced kidney size/growth, resulting in an 'accumulation' of the stromal progenitor population that may be unable to differentiate into the

various populations interstitium due to a lack of signals from normally developing nephrons. However, ALDH1A2 expression at E18.5 suggests that this apparent 'cell-autonomous expansion' does not continue at later time points in development, as the quantification of the nephrogenic zone stroma is significantly decreased at E18.5, with only minimal interval growth of the overall kidney size over this period. Taken together, findings from the *Six2cre;RosaDTA*$^{c/+}$ model show that the expansion of the *Foxd1*⁺ stromal progenitor population at E15.5 occurs independent of the signals from the nephrogenic niche and overall kidney size that interestingly is not maintained later in development (i.e. E18.5), suggesting that dynamic, time-dependent mechanisms regulate the stromal progenitor population throughout development.

### *Foxd1*⁺ progenitor cells from *Six2cre;Wt1*$^{c/c}$ mutant kidneys show transcriptional changes in addition to an abnormal expansion

Given the abnormal expansion of nephrogenic zone stroma identified in E15.5 mutant kidneys, we next sought to evaluate for differences in transcriptional regulation of the *Foxd1*⁺ progenitor population in *Six2cre;Wt1*$^{c/c}$ mutant model versus *Six2cre* controls. As shown in Fig. 2, unsupervised clustering identified eight stromal clusters, with enriched expression of nephrogenic zone stromal markers in clusters 14, 1, and 7 in both control and mutant kidneys (Fig. 5A). To evaluate for differences in the nephrogenic zone stroma, we first sought to compare changes in the *Foxd1*⁺ population by combining these three clusters together and evaluating for DEGs filtered on log2FC>0.2 and FDR adjusted *P*-value <0.05 (Fig. 5B-O; Table S4). A number of genes, including *Ntn1*, *Ncam1* and *Aldh1a2*, show no statistical differences in expression (Fig. 5B,F,L). However, other genes, including *Foxd1*, *Ebf3*, *Ncam2*, *Cdh2*, *Rmst*, *Chl1* and *Nrxn1* show increased expression in the nephrogenic zone stroma, with gene expression across all stromal clusters shown via UMAP (Fig. 5C,D,G, H,I,J,M). Of note, given that the NCAM antibody used for IF (Fig. 1M-P) recognizes both NCAM1 and NCAM2, the increased *Ncam2* expression identified by snRNA-seq likely accounts for this difference in IF staining. Next, we evaluated for DEGs in each of the subclusters of *Foxd1*⁺ progenitor cells, as shown by dot plots for DEGs in cluster 14, 1 and 7 (Fig. 5P). Selected genes filtered on log2FC>0.2 and FDR adjusted *P*-value <0.05 showing increased expression in mutant nuclei are highlighted in red and those showing decreased expression are highlighted in blue (Fig. 5P; Table S4). Thus, snRNA-seq identifies a number of differentially regulated genes, both in individual subclusters as well as across the *Foxd1*⁺ progenitor population, suggesting that these cells not only show an abnormal expansion but also demonstrate transcriptional changes in response to defects in nephrogenesis in the *Six2cre;Wt1*$^{c/c}$ model.

### Heterogeneity identified in the *Foxd1*⁺ stromal population of murine kidneys, including markers from cluster 14, appear to be conserved in the developing human fetal kidney

While comparative studies have shown that many features of kidney development are conserved between mice and humans, recent studies have identified a number of divergent features and gene expression differences (Kim et al., 2024; Lindström et al., 2018a,b). To evaluate if the markers identified in murine stromal progenitor cells are conserved in human development, we re-analyzed publicly available human fetal kidney snRNA-seq data from 10.6 to 17.6 weeks gestation (Kim et al., 2024). Unsupervised cell clustering identified a total of 31 clusters, with 26 corresponding to various cell types based on marker gene lists and anchor gene expression (Fig. 6A), with five clusters excluded (i.e. clusters 22,

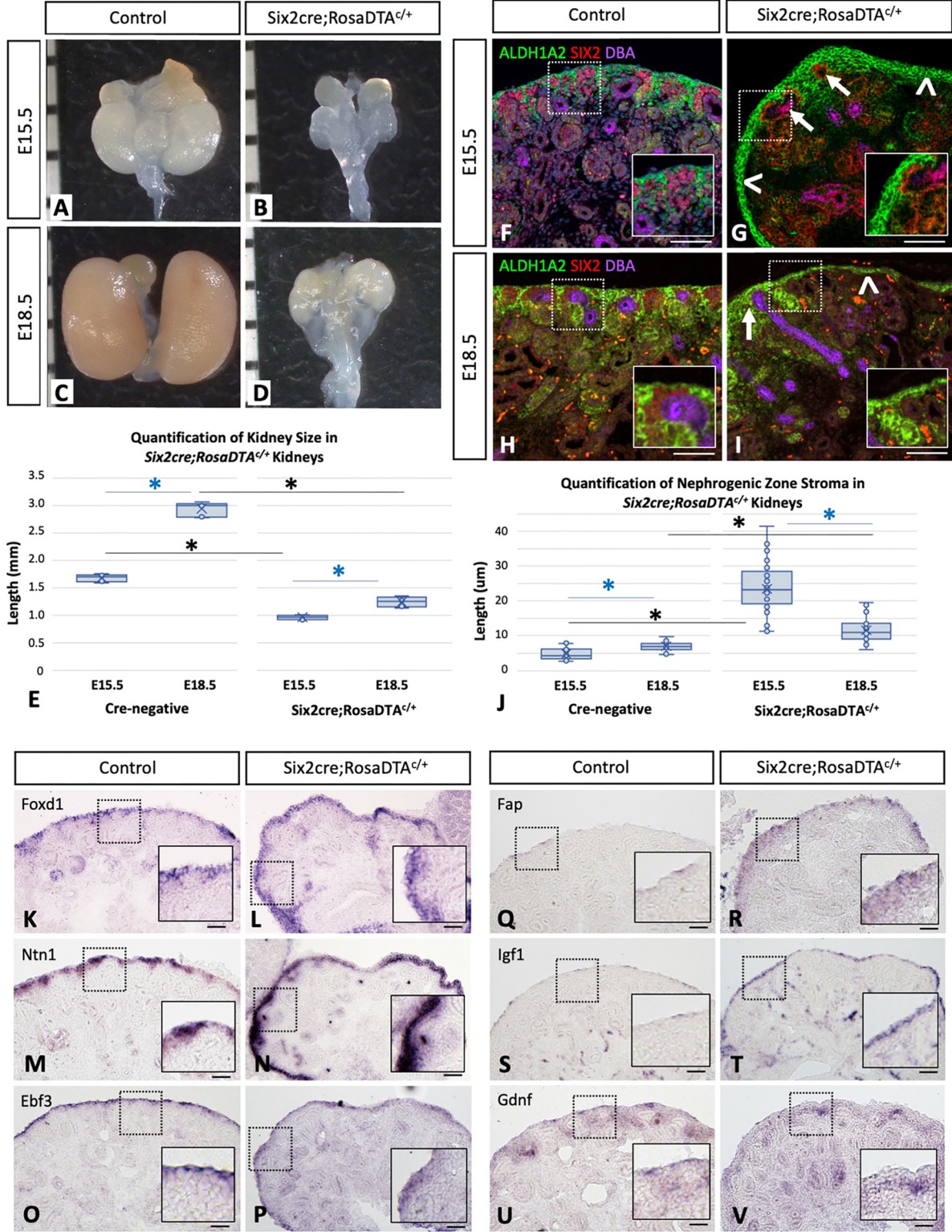

**Fig. 4. Stromal progenitor cells are maintained independent of the nephrogenic niche in *Six2cre;RosaDTA^{c/+}* mutant kidneys.** (A-E) *Six2cre; RosaDTA^{c/+}* mutant kidneys show severe defects in nephrogenesis resulting in smaller kidneys at E15.5 and E18.5 (B and D, respectively) compared to cre-negative littermate controls (A and C, respectively) quantified by kidney length (E; paired *t*-test in blue and unpaired *t*-test in black; asterisks denote *P*-value of ≤0.001). (F-J) Evaluation of the nephrogenic zone stroma at these time points shows that the abnormal expansion at E15.5 (G) appears to improve/resolve by E18.5 (I), which was quantified by measuring the distance from the outer cap mesenchyme to the periphery of the kidney on sections of control and mutant kidneys stained with ALDH1A2 (J; paired *t*-test in blue and unpaired *t*-test in black; asterisks denote *P*-value of ≤0.001). Box plots show data points (inner and outlier points) with median line and first to third interquartile ranges; whiskers indicate 1.5× the interquartile ranges. (K-V) *In-situ* hybridization (ISH) of *Foxd1*, *Ntn1*, *Ebf3*, *Fap*, *Igf1* and *Gdnf* shows an expression pattern consistent with subcluster 14 of the *Foxd1*⁺ progenitors in the snRNA-seq dataset, suggesting that these cells are maintained even in regions lacking NPCs/UB. E15.5 and E18.5 representative images shown from *n*=3 with three separate embryos used from both control and mutant kidneys for quantification, whole kidney images, IF and ISH. Scale bars: 50 µm.

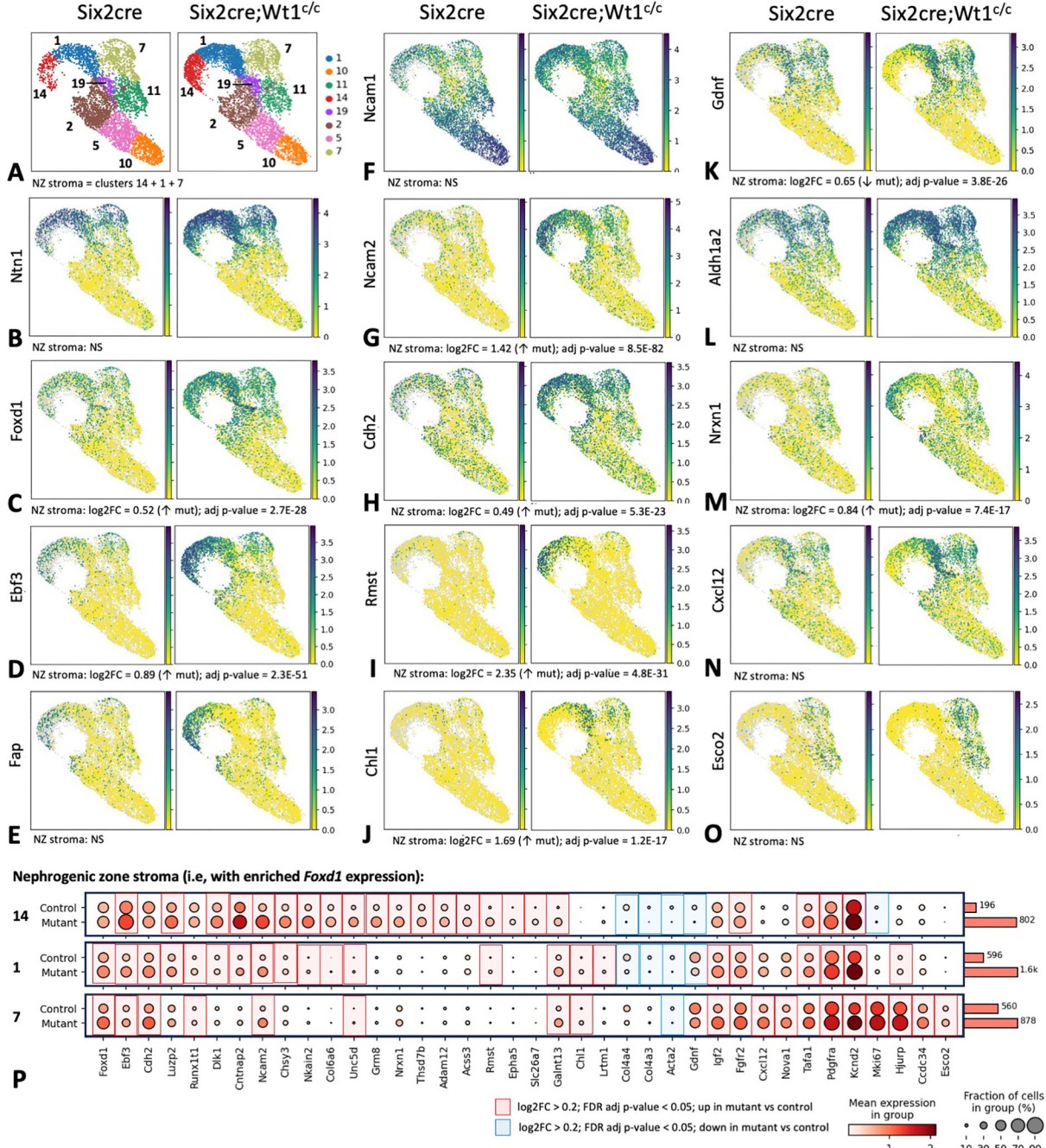

**Fig. 5. SnRNA-seq of control (*Six2cre*) and mutant (*Six2cre;Wt1^{c/c}*) kidneys identifies differentially expressed genes in the *Foxd1*⁺ stromal progenitor population.** (A) Unsupervised clustering of control and mutant embryonic kidney stromal cells identifies eight clusters, three of which (14, 1 and 7) show enriched expression of nephrogenic zone stromal markers, including *Foxd1*, as shown in Fig. 2 and here in panel C. (B-O) To evaluate for differentially expressed genes (DEGs), control and mutant *Foxd1*⁺ progenitor cells (i.e. clusters 14, 1 and 7, labeled as nephrogenic zone or 'NZ' stroma) were pooled and compared, with log2FC and FDR adjusted *P*-value calculated for only the NZ stromal clusters and UMAP showing gene expression across all stromal clusters. Several genes, including *Ntn1*, *Ncam1* and *Aldh1a2*, show no statistical differences in expression (B,F,L, respectively). However, other genes, including *Foxd1*, *Ebf3* and *Ncam2*, show increased expression in the *Foxd1*⁺ stroma of *Six2cre;Wt1^{c/c}* kidneys (C,D,G, respectively). (P) To evaluate for differential expression across the distinct subclusters of *Foxd1*⁺ cells, dot plots were generated for DEGs in the three separate clusters. Genes filtered on log2FC>0.2 and FDR adjusted *P*-value<0.05 showing increased expression in mutant nuclei are highlighted in red and downregulated are highlighted in blue, with this analysis showing a number of upregulated genes suggesting abnormal expression in the stromal progenitor population of mutant kidneys.

24, 28, 29 and 30) due to either relatively low reads, high ribosomal expression, and/or lack of specific marker expression (Table S5). Turning our focus to the stroma, we identified six subpopulations (i.e. clusters 1, 2, 6, 11, 12 and 23) totaling 11,744 nuclei. Three clusters (i.e. 1, 11 and 12) showed enriched expression of the

nephrogenic zone stromal markers *NTN1* and *FOXD1* (though overall low levels of *FOXD1* expression limited the utility of this marker in this dataset), with cluster 12 showing increased expression of proliferation markers *TOP2A* and *MKI67*, suggesting this cluster reflects a cycling state. Marker gene

analysis additionally identified expression of *COL14A1* and *MGAT4C* specific to cluster 1 versus *MYOCD* and *FST* expression enriched in cluster 11. The other stromal clusters (i.e. clusters 2, 23 and 6) showed enriched expression of cortical and medullary stromal markers including *GUCY1A1*, *COL23A1*, *APCDD1* and *ALX1* (Fig. 6B; Table S5). Additional analyses of snRNA-seq marker genes and DEGs via heatmaps for the murine and human stromal clusters (Fig. S5) support that the identified stromal clusters show distinct transcriptional profiles suggestive of diverse cell types.

Next, we sought to compare the expression of specific stromal markers in the mice versus human fetal kidney snRNA-seq datasets. To do this, we generated dot plots of stromal markers showing their expression across the identified stromal clusters (Fig. 6C), with conserved genes highlighted in green, divergent genes highlighted in red and murine-specific genes highlighted in gray. Focusing on the markers expressed in cluster 14 of the *Foxd1*+ progenitor cells from the murine snRNA-seq dataset, we observed similarly enriched expression of a number of markers in cluster 1 of the human fetal kidney snRNA-seq, including *EBF3*, *LUZP2*, *COL6A6*, *PRRX1*, *IGF1*, *DKK2*, *RXFP2*, *FAP*, *RSPO3*, *SMOC2* and *POSTN*, with low *GDNF* expression (Fig. 6C). However, other murine markers, including *Aldh1a2*, *Cntnap2*, *Gria1*, *Dlk1*, *Grm8*, *Ptger3*, *Shisa9* and *Aox3*, showed divergent expression patterns between the murine and human datasets. Nonetheless, the number of shared genes showing specifically enriched expression in cluster 14 of the murine dataset and cluster 1 of the human dataset suggest that this stromal progenitor subcluster is likely conserved in the developing kidneys of both mice and humans, and that this subcluster may represent a unique subset of cells within the stromal progenitor population.

## DISCUSSION

While it has long been recognized that interactions among the NPCs, stromal progenitor cells and UB play crucial roles in regulating kidney development, how cell-autonomous versus cell-lineage crosstalk regulates various events in normal development is still being understood. Here, we show that mouse models with defects in nephrogenesis (i.e. *Six2cre;Wt1^c/c^*, *Wnt4-null* and *Six2cre;RosaDTA^c/+^*) demonstrate an abnormal expansion of the *Foxd1*+ stroma at E15.5, as summarized in Fig. 7. Additionally, the *Six2cre;RosaDTA^c/+^* model shows that the expansion of *Foxd1*+ progenitor cells occurs independent of signals from the nephrogenic niche that is interestingly not sustained at later stages (i.e. E18.5), suggesting cell-autonomous regulation of the *Foxd1*+ stromal progenitor population that varies with developmental stage/time. Furthermore, snRNA-seq analyses identify significant transcriptional changes in the stromal progenitor population of the *Six2cre;Wt1^c/c^* and confirm its expansion at E15.5, suggesting that both cellular and molecular dysregulation of the *Foxd1*+ progenitor cells occurs in response to defects in nephrogenesis.

### The *Foxd1*+ stromal progenitor population is heterogeneous, with a subset of cells showing a distinct transcriptomic profile that appears to be conserved in human fetal kidneys, and undergoes an abnormal expansion in mouse models with defects in nephrogenesis

SnRNA-seq analysis of the *Foxd1*+ stroma from control and *Six2cre;Wt1^c/c^* mutant kidneys at E15.5 identifies three subclusters, including a distinct subcluster localized to the outer periphery of the nephrogenic zone stroma. Interestingly, this specific subset of the *Foxd1*+ stromal progenitor population appears to be expanded in

*Six2cre;Wt1^c/c^*, *Wnt4-null* and *Six2cre;RosaDTA^c/+^* mutant kidneys by ISH validation of several markers. Additionally, this expansion appears to be independent of signals from the NPCs/UB in the nephrogenic niche, as per our analyses of the *Six2cre;RosaDTA^c/+^* mutant model. This is also similar to what occurs to the NPCs in the stromal-ablation model, as apoptosis of the stromal progenitor cells with diphtheria toxin (i.e. *Foxd1cre;RosaDTA^c/+^* mutant kidneys) results in maintained self-renewing NPCs that fail to differentiate and actually show an 'expansion' of their population due to the loss of signals from the developing stroma (Das et al., 2013; Hum et al., 2014). These findings thus suggest that neither the nephron nor stromal progenitor populations rely on signals from one another for their maintenance throughout development.

Given the global defects in the mutant mouse models, we sought to evaluate how a lack of kidney growth and stunted development may confound the observed phenotype. To partially address this, we evaluated the *Six2cre;RosaDTA^c/+^* model for kidney growth and changes in nephrogenic stroma over time at E15.5 and E18.5, as shown in Fig. 4. While our data suggest that the stromal progenitor population is slightly expanded at E13.5 compared to E15.5 in normally developing kidneys (Fig. 1V), the *Six2cre;RosaDTA^c/+^* model reveals a marked expansion of the nephrogenic zone stroma at E15.5 that is not present at E18.5 (Fig. 4J). This suggests that continued proliferation/expansion of the stromal progenitor population occurs at early time points in development, regardless of signals from the NPCs/UB and regardless of kidney size, which may result in an 'accumulation' of the stromal progenitor population unable to differentiate due to a lack of signals from normally developing nephrons. However, the lack of continued or increasing expansion of stromal progenitor cells at later developmental time points (i.e. E18.5) in *Six2cre;RosaDTA^c/+^* mutant kidneys suggests dynamic, time-dependent changes in the *Foxd1*+ population that may be reflective of changes in 'young versus old' stromal progenitor cells that may be evaluated in future studies.

Additionally, it is important to consider whether abnormal signaling from the UB may be another possible mechanism underlying disruptions in stromal development in the three mutant kidney models. To partially address whether effects of decreased UB branching and a 'delayed' developmental phenotype may contribute to the observed phenotype, we evaluated the expression of *Foxd1*+ stromal markers at E12.5 and E13.5. These findings show that the abnormal expansion of *Foxd1* and *Ntn1* expression in mutant kidneys is not solely due to decreased UB branching, as it is not seen in E12.5 or E13.5 kidneys when the kidneys are smaller with less branching. However, it is still possible that signaling from abnormally developed UBs (i.e. either from the inability to branch normally or from altered development due to disrupted signaling from the NPC lineage) may contribute to the disrupted stromal development in the mutant kidneys. Despite this limitation of the study, the findings in this report nonetheless provide additional insights into the regulation of the *Foxd1*+ stromal progenitor population.

While this report sought to comprehensively evaluate stromal development mouse models with defects in nephrogenesis, one previous study examining ablation of *Mdm2* targeted to the nephron lineage also reported 'multi-layering' of stroma at the outer periphery of the developing kidney. This model resulted in defective nephrogenesis via NPC apoptosis resulting in a 'thinning of the cap mesenchyme' in mutant kidneys (Hilliard et al., 2014), raising the possibility that stromal progenitor cells may 'fill the void' of lost NPCs. However, our snRNA-seq dataset shows stromal progenitor expansion in *Six2cre;Wt1^c/c^* kidneys without a deficit in self-renewing NPCs (Table S3; Fig. S3). Additionally, the authors of that study also

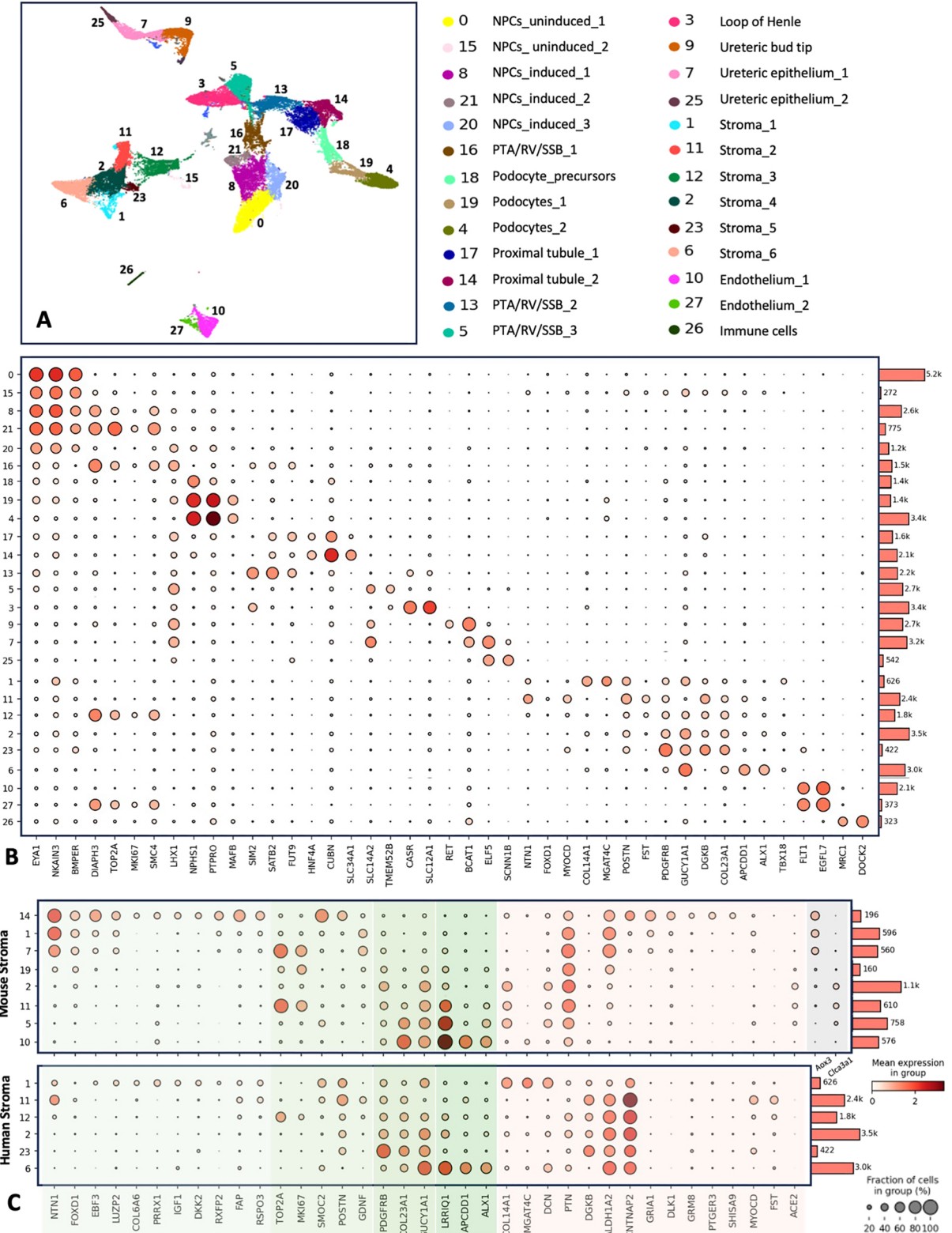

**Fig. 6. Human fetal kidney snRNA-seq identifies distinct stromal cell types.** (A) Publicly available snRNA-seq data, including 52,046 nuclei from 10.6 to 17.6 weeks gestation human fetal kidneys, were re-analyzed to specifically evaluate subclusters of human fetal kidney stroma. (B) Cell type-specific markers were used to label cell types after performing unsupervised clustering, which identified a total of 27 clusters corresponding to specific cell types, including six clusters of stroma (i.e. 1, 2, 6, 11, 12, and 23) showing distinct expression profiles. (C) To evaluate how transcriptional markers from murine stromal cell clusters compare to those identified in the human fetal kidney dataset, marker gene expression was compared across the stromal clusters from both the mouse and human datasets, with similarly expressed genes highlighted in green, divergent genes highlighted in pink, and two markers specific to mouse stroma and not expressed in humans highlighted in gray. Notably, stromal subcluster 1 in the human dataset shows conserved expression of *NTN1*, *EBF3*, *LUZP2*, *COL6A6*, *PRRX1*, *IGF1*, *DKK2*, *RXPF2*, *FAP* and *RSPO3* as well as a low expression of *GDNF* similar to subcluster 14 in mice, suggesting that this subpopulation of the *Foxd1*+ stromal progenitor cells may be conserved between the species.

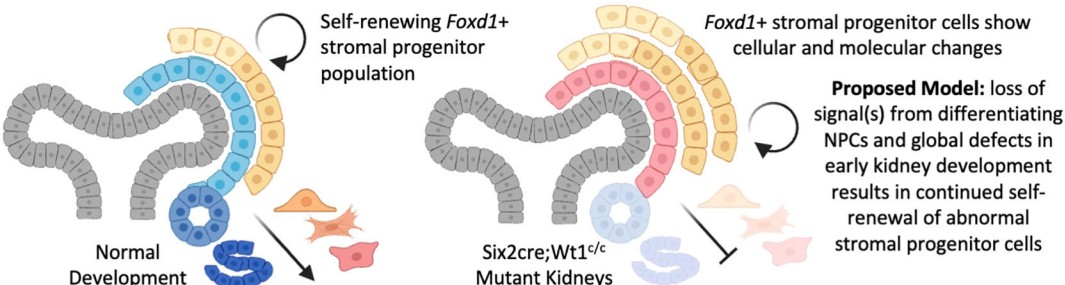

**Fig. 7. Proposed model for the stromal progenitor expansion in mutant kidneys with defects in nephrogenesis.** Findings from three mouse mutant models (*Six2cre;Wt1^{c/c}*, *Wnt4-null*, *Six2cre;RosaDTA^{cl/+}*) suggest that disruptions in nephron progenitor differentiation and global defects in development with smaller kidney size result in an abnormal expansion and gene expression changes of the *Foxd1^+* stromal progenitor population.

reported that IF of proliferating cell nuclear antigen (PCNA) suggested increased proliferation, which is not clearly consistent with our *Six2cre;Wt1^{c/c}* snRNA-seq data. While further experiments will be necessary to conclude if the expansion of the stromal progenitor population reported here may result from either abnormally increased self-renewal/proliferation or a block in stromal differentiation leading to an accumulation of undifferentiated stromal progenitor cells, or both, these findings nonetheless demonstrate abnormal regulation of the stromal progenitor population that occurs due to defects in nephrogenesis.

## Implications of stromal progenitor regulation in normal kidney development and disease

The nephrogenic niche consists of a specialized microenvironment that has been shown to regulate the balance of progenitor cell maintenance/differentiation in coordinating nephrogenesis. The *Foxd1^+* stromal progenitor population localized to this niche is known to play a crucial role in establishing this microenvironment and gives rise to interstitial cells necessary for normal kidney development (Wilson and Little, 2021) shown to be important in response to injury (Muthukrishnan et al., 2018). Here, we show that defects in nephrogenesis, including those caused by mutations in *Wt1*, result in abnormal stromal progenitor cell expansion and transcriptional changes. This is of particular interest given that loss-of-function of WT1 is associated with a number of renal pathologies, including Denys Drash syndrome, WAGR (Wilms tumor, aniridia, genitourinary anomalies, and a range of developmental delays) and predisposition to Wilms tumor (Torban and Goodyer, 2024). Given that the nephrogenic zone stroma is known to provide signals to the NPCs, its abnormal expansion and transcriptional changes may potentially cause aberrant crosstalk within the nephrogenic niche, which may be of interest to explore in future studies. For example, *Six2cre;Wt1^{c/c}* mutants show an increase of *Cxcl12* in the proliferating cluster of the *Foxd1^+* population, with a previous publication suggesting that increased stromal expression of *Cxcl12* inhibits UB branching and nephron endowment (D'Cruz et al., 2023). While additional studies are necessary to uncover the complex signaling within the nephrogenic niche that may be disrupted when NPCs fail to differentiate and are also needed to validate the identified stromal progenitor markers in human fetal kidneys and/or kidneys affected by WT1-related diseases, this report identifies a distinct subset of the multipotent stromal progenitor cells within the *Foxd1^+* population that shows cellular and molecular changes with defects in nephrogenesis, thus offering the opportunity to develop novel tools and uncover additional mechanisms in studying the role of the stroma in development and disease.

## MATERIALS AND METHODS
### Animal models

All animals were housed, maintained and used according to National Institutes of Health (NIH) and Institutional Animal Care and Use Committees (IACUC) approved protocols at the University of Texas Southwestern Medical Center (OLAW Assurance Number D16-00296). Mouse lines used in this study were obtained from The Jackson Laboratory and include: *Six2creTGC*, Jax strain #009606 (Kobayashi et al., 2008); *Wt1^{c/c}*, Jax strain #019554 (Gao et al., 2006); *RosaDTA^{c/c}*, Jax strain #006331 (Brockschnieder et al., 2004); and *Wnt4-null*, Jax strain #002866 (Stark et al., 1994). All mice were bred on a mixed genetic background. For experimental assays, females homozygous for conditional alleles were crossed with male cre-line mice, with the day of plug counted as E0.5. Pregnant females were sacrificed at various gestational time points. Lineage-tracing experiments were performed by crossing *Rosa26^{EYFP}* (Jax strain #006148; Srinivas et al., 2001) reporter mice with the above mouse lines. Mice with the desired genotype were randomly selected regardless of sex. Cre-negative littermates were used as controls.

### Kidney sample preparation, histology, immunofluorescence and *in-situ* hybridization

Embryonic tissue was fixed in 4% paraformaldehyde (PFA), embedded in paraffin, sectioned into 5 μm slices and subjected to H&E staining or IF. Slides for IF were immersed and boiled with either 10 mM sodium citrate or TE antigen retrieval buffer and blocked with a solution of 5% normal donkey serum for 1 h at room temperature followed by the application of primary antibodies diluted in blocking solution. The following primary antibodies were used: SIX2 (Proteintech; 11562-1-AP; 1:200; rabbit; RRID: AB_2189084), SIX2 (Abnova; H00010736-M01; 1:200; mouse; AB_436993), LXH1 (Developmental Studies Hybridoma Bank; 4F2; 1:100; mouse; RRID: AB_531784), NCAM (Sigma-Aldrich; C9672; 1:200; mouse; AB_1079450; recognizes both NCAM1 and NCAM2), CK (DSHB; Troma-III; 1:50; rat; AB_2133570), MEIS 1/2/3 (Active Motif; 39795; 1:100; mouse; RRID: AB_2143020); ALDH1A2 (Sigma-Aldrich; HPA010022; 1:200; rabbit; RRID: AB_1844723), DBA (Vector; B-1035; 1:500; Biotinylated; RRID: AB_2314288), GFP (Aves; GFP-1020; 1:200; chicken; RRID: AB_10000240), PCNA (Abcam; ab18197; 1:200; rabbit; RRID: AB_444313) and pHH3 (Sigma-Aldrich; H0412, 1:500; rabbit; RRID: AB_477043). IF staining of control and mutant paraffin sections was carried out on the same slide, and visualization was carried out using the same microscope/photography settings (NikonA1 inverted confocal microscope). For ISH assays, anti-sense RNA digoxigenin-labeled probes were generated from cDNA from commercially available plasmids from Horizon including: *Ntn1* (MMM1013-202706854), *Ebf3* (MMM1013-202859613), *Aox3* (MMM1013-202708447), *Fap* (MMM1013-202767860), *Igf1* (MMM1013-202767860), *Gria1* (MMM1013-202859754), *Gdnf* (MMM1013-211692457) and *Ptn* (MMM1013-211692457). The *Foxd1* plasmid was provided by the Carroll lab. Kidney tissue was fixed with 4% PFA, cryoprotected with 30% sucrose, embedded in OCT medium (TissueTek), sectioned into 10 μm sections and rehydrated with PBS before being treated with 15 μg/ml proteinase K for 10 min and

fixed in 4% PFA followed by an acetylation step. Slides were then washed and incubated with pre-hybridization buffer for 1 h at room temperature before being hybridized with the specific probe overnight at 65°C. Slides were then washed in 0.8× saline sodium citrate (SSC) then transferred to sodium chloride, Tris and Tween-20 (NTT) before blocking with 2% blocking solution (Roche) for at least 1 h at room temperature. Slides were then incubated with anti-Dig alkaline phosphatase-conjugated antibody (Roche, 1:4000) overnight at 4°C. The next day, slides were washed three times in NTT and three times in sodium chloride, Tris, Tween-20, magnesium chloride and levamisole (NTTML) before incubating with BM purple (Roche); slides were then fixed with 4% PFA and mounted using glycergel (Dako).

### Single nuclei RNA-seq of E15.5 murine kidneys

Nuclei were isolated as previously reported (Wu et al., 2018) from paired kidneys harvested from E15.5 control and mutant embryos with Nuclei EZ Lysis buffer (Sigma-Aldrich, NUC-101) supplemented with protease inhibitor (Roche, 5892791001) and RNase inhibitor (Promega, N2615 and Life Technologies, AM2696). Samples were homogenized using a Dounce homogenizer (Kimble Chase, 885302-0002) in 2 ml of ice-cold Nuclei EZ Lysis buffer first with a loose pestle, then passed through a 200 µm strainer into a 50 ml conical tube and followed by homogenization with a tight pestle which was then filtered through a 40-µm cell strainer (pluriSelect, 43- 50040-51) and incubated on ice for 5 min. The suspension was centrifuged at 500 *g* for 5 min at 4°C, with the pellet resuspended and washed with 4 ml of the buffer and incubated on ice for 5 min. After another centrifugation, the pellet was resuspended with Nuclei Suspension Buffer (1× PBS, 0.07% bovine serum albumin, 0.1% RNase inhibitor) and filtered through a 5-µm cell strainer (pluriSelect, 43-50020-50) into a conical tube. The nuclei were counted using a disposable hemocytometer and diluted to a concentration of 700-1200 cells per µl. Nuclei were sequenced using the 10x Chromium Single Cell Platform (10x Genomics) targeting 5000 nuclei per sample and 50,000 reads per nuclei. Library preparation and sequencing was completed by the UT Southwestern McDermott Sequencing core using the 3′ GEX kit on NextSeq 2000 (5-150 cycle high output) and aligned to GRCm39.

### Bioinformatic analyses of single nuclei RNA-seq

10x Genomics data were analyzed using the Cell Ranger Pipeline cellranger (Version 5.0.1) for alignment, quantification and initial preprocessing of single nuclei RNA-seq data. The reads were aligned using STAR to the mouse reference genome provided by 10x Genomics (refdata-gex-mm10-2020-A). The estimated cell number was derived by plotting the UMI counts against the barcodes. Further filtering of the expression data matrices was carried out to ensure high quality data. Cells with a minimum number of UMI≥500, a minimum number of detected genes≥250, log10GenesPerUMI>0.80 and mitoRatio<0.20, were selected. Cells were clustered using the Louvain algorithm implemented in Seurat based on the top principal components (Butler et al., 2018; Hao et al., 2021; Satija et al., 2015; Stuart et al., 2019). Both Seurat (4.0.5) and CZ CellxGene (1.3.1) (https://doi.org/10.1101/2021.04.05.438318; https://www.biorxiv.org/content/10.1101/2020.08.28.270652v2) were used for clustering analysis, differential expression analysis and visualization of single cell RNA-seq data. Cell clusters were identified based on marker gene expression and visual inspection (Chaney et al., 2022). SnRNA-seq from three separate samples (i.e. paired kidneys from one cre-negative and two separate *Six2cre* embryos and three separate *Six2cre;Wt1^{c/c}* embryos) were used. Differential abundance testing was conducted with the R package Milo (Dann et al., 2022). Neighborhoods with more than 75% of member cells belonging to a single identity were assigned that identity; otherwise, they were assigned the identity 'Mixed'.

Publicly available human fetal kidney snRNA-seq data was re-processed using Seurat (version 4.3.0). Raw counts were log-transformed and the top 4000 variable genes were selected for scaling, principal component analysis and batch correction with Harmony (version 1.2.0). The top 20 dimensions were selected for downstream processing, including UMAP and Louvain clustering. Clusters were calculated at multiple resolutions and an optimal resolution was selected based on examination of a cluster tree (version 0.5.1) plot and marker gene lists.

### Statistical analysis and data visualization

Statistics for the bioinformatic analyses of the single nuclei RNA-seq data included: statistical significance of DEG lists of control and mutant determined by Wilcoxon Rank-Sum Test with significance thresholds adjusted for multiple testing (<0.05) with the data filtering/analyses as described above. Data presented in the figures are representative images from one of at least three different experiments on different embryos/organs. No significant variability was noted in tissues of the same genotype; all animals with correct genotypes were included in the analysis. ImageJ was used to measure nephrogenic stromal width on immunofluorescence images stained with ALDH1A2 by measuring the distance from the outer cap mesenchyme (i.e. SIX2⁻ cells) to the periphery of the kidney on sections of control and mutant kidneys. Ten measurements were obtained from three separate embryos of each of the samples/time points analyzed. Measurements from *Six2cre;RosaDTA^{c/+}* mutants included an equal number of measurements each from regions where the NPCs were present and regions in which they were completely ablated. Statistical significance evaluated by one way ANOVA and Tukey HSD was used for group comparisons, with paired two-tailed *t*-test used for related groups (i.e. E15.5 versus E18.5 time points from the same genotype) and unpaired two-tailed *t*-test for independent group comparisons (i.e. different genotypes).

### Acknowledgements

This research was supported in part by the computational resources provided by the BioHPC supercomputing facility located in the Lyda Hill Department of Bioinformatics, UT Southwestern Medical Center, TX, USA (https://portal.biohpc.swmed.edu).

### Competing interests

A.P.M. is a consultant or scientific advisor to Novartis, eGENESIS, Trestle Biotherapeutics and IVIVA Medical. All other authors declare no conflict of interest.

### Author contributions

Conceptualization: K.A.D.; Data curation: M.G.M., Y.M., D.G., S.K., A.P.M., K.A.D.; Formal analysis: Y.L., D.R.R., J.T.L., Y.M., C.X., K.A.D.; Funding acquisition: K.A.D.; Methodology: J.T.L., Y.M., T.J.C., K.A.D.; Software: J.T.L.; Writing – original draft: M.G.M., K.A.D.; Writing – review & editing: J.T.L., A.P.M., T.J.C., K.A.D.

### Funding

This work was supported by the National Institute of Diabetes and Digestive and Kidney Diseases (NIDDK) grants K08DK131258 (to K.A.D.), RC2DK125960 and P30DK079328 (to T.J.C.), with work in A.P.M.'s laboratory supported by NIDDK DK054364 and the Chan Zuckerberg Initiative WU-20-101 as part of the Seed Network of the Human Cell Atlas consortium (HCA). Open Access funding provided by University of Texas Southwestern Medical Center. Deposited in PMC for immediate release.

### Data and resource availability

SnRNA-seq data from mouse control and *Six2cre;Wt1^{c/c}* mutant kidneys generated in this study (including raw and processed files) have been deposited in GEO under accession number GSE289696. Human fetal kidney snRNA-seq is available from GEO (GSE232479) as previously published (Kim et al., 2024). All other relevant data and details of resources can be found within the article and its supplementary information.

### Peer review history

The peer review history is available online at https://journals.biologists.com/dev/lookup/doi/10.1242/dev.204964.reviewer-comments.pdf

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
