## [Peer Review File · Development (Cambridge, England)]

Defects in nephrogenesis result in an expansion of the *Foxd1*⁺ stromal progenitor population

Michael G. Michalopoulos, Yan Liu, Dinesh Ravindra Raju, John T. Labin, Yanru Ma, Dhruv Gaur, Sadiksha Khadka, Chao Xing, Andrew P. McMahon, Thomas J. Carroll and Keri A. Drake

DOI: 10.1242/dev.204964

Editor: James Wells

Review timeline

Original submission:	21 May 2025
Editorial decision:	2 July 2025
First revision received:	6 October 2025
Accepted:	5 November 2025

Original submission

First decision letter

MS ID#: dev.204964

MS TITLE: Defects in nephrogenesis result in an expansion of the *Foxd1*⁺ stromal progenitor population

AUTHORS: Michael G. Michalopoulos, Yan Liu, Dinesh Ravindra Raju, John T. Labin, Yanru Ma, Dhruv Gaur, Sadiksha Khadka, Chao Xing, Andrew P. McMahon, Thomas J. Carroll and Keri Anne Drake

Dear Dr Drake,

We apologize for the delay in the review process. As we have been unable to get the third reviewer to finalize their critique I am proceeding with a decision using the two reviews. The referees' comments are appended below, or you can access them online: please go to:

As you will see, the referees express considerable interest in your work, but have some significant criticisms and recommend a substantial revision of your manuscript before we can consider publication. If you are able to revise the manuscript along the lines suggested, which may involve further experiments, I will be happy receive a revised version of the manuscript. Your revised paper will be re-reviewed by one or more of the original referees, and acceptance of your manuscript will depend on your addressing satisfactorily the reviewers' major concerns. Please also note that Development will normally permit only one round of major revision. If it would be helpful, you are welcome to contact us to discuss your revision in greater detail. Please send us a point-by-point response indicating your plans for addressing the referees' comments, and we will look over this and provide further guidance.

Please attend to all of the reviewers' comments and ensure that you clearly highlight all changes made in the revised manuscript. Please avoid using 'Tracked changes' in Word files as these are lost in PDF conversion. I should be grateful if you would also provide a point-by-point response detailing how you have dealt with the points raised by the reviewers in the 'Response to Reviewers' box. If you do not agree with any of their criticisms or suggestions please explain clearly why this is so.

Reviewer 1*Advance summary and potential significance to field*

This paper aims to understand how signals from the developing nephrogenic lineage regulate stromal development. To address this question, three mutant mouse lines were generated: Six2Cre-Wt1 cKO, loss of Wnt4 which is a downstream effector of Wt1 in regulation of nephrogenesis, and NP ablation by Six2Cre-DTA, each introducing disruptions in nephrogenic lineage development. Wt1 is known to maintain the metanephric mesenchyme and drive the nephrogenic process in the embryonic kidney. Loss of Wt1 results in widespread apoptosis in developing kidney and eventually kidney agenesis. Mutated WT1 is related to formation of Wilm's Tumor. In this study, all three mutant mouse lines exhibited anomalies in nephron progenitors and nephrogenic intermediates, to a variable extent, and demonstrated an expanded cell population located at the outer periphery of kidneys in the place of stromal progenitors. Age-matched control kidneys were then analyzed by single-nuclei RNA sequencing, which identified subpopulations of stromal progenitors. Analyses in mutant kidneys then confirmed that the expanded peripheral population expresses some identified markers of stromal progenitors, and that this expanded population is not due to loss of nephron progenitor cells. The authors then analyzed three Wilm's Tumor samples by single-nuclei RNA sequencing to compare with control human fetal kidneys and control mouse kidneys, and discovered that the majority of stromal cells from Wilm's Tumor samples are transcriptionally similar to stromal progenitors.

Comments for the author

Major comments:

1. In the absence of a more comprehensive analysis of stromal cells in mutant kidneys and their gene expression, one does not know what the expanded population represents. That is, does it represent an increased number of normal stromal cells, or does it represent an unusual population of stromal cells? While some stromal progenitor markers are expressed in the expanded population, it is not very likely that this population simply results from increased normal stromal cells, since (1) NCAM is ectopically expressed in this population, and (2) cell proliferation is unchanged in the stroma of mutant kidneys.
2. While this study demonstrates that an abnormal nephrogenic lineage can affect the stromal lineage, it did not reveal the mechanisms underlying this interaction. What is the nature of this non-cell-autonomous regulation? How do abnormalities in the nephrogenic lineage affect stromal development?
3. To address these questions above, one would like to see the transcriptome of the mutant kidneys being analyzed at a single-cell level, with special focus on stromal cells, to compare with the presented control data. This experiment is necessary given that these questions are at the heart of the study.
4. While the comparison between Wilm's Tumor samples and healthy control kidneys is interesting, this part alone is not very helpful in understanding the abnormal interaction between nephrogenic cells and stromal cells demonstrated by mutant kidneys, as the connection between tumor samples and mutant mouse kidneys remains questionable until the anomalies in the mutant mouse kidneys have been fully characterized.
5. It is understood that the tumor samples have some similarities to mutant kidneys in terms of expanded stroma. However, only one patient sample was identified as positive for WT1 mutation, and the one that exhibits nephrogenic rests is negative for a WT1 mutation. Moreover, unlike patients that carry global mutations, mutant mice carry lineage-specific mutations in the nephrogenic lineage. How does this WT1 variant affect the function of WT1 at molecular level? Does it lead to a complete loss of function similar to Cre deletion? To what degree do the mutant mouse kidneys and human Wilm's Tumor mimic each other?
6. Line 151-153: One would ask: are these cells really derived from stromal progenitors? Couldn't it be maldeveloped nephrogenic cells that gained some stromal gene expression moving outwards, given that they are NCAM+? The YFP lineage tracing (line 146-148) is very helpful in addressing this issue but this needs to be explicitly discussed.
7. Line 362-363: data integration can mask ectopic population as this step forcefully maps populations between datasets. It is understood that correction is inevitable before combining datasets but this limitation needs to be discussed.

Minor comments:

1. One would like to see confirmation of loss of Wt1.
2. Line 130-131: please elaborate the anomalies in nephrogenic structures. While LHX1+ structures are absent, it seems that NCAM1 is present in all three mutants.
3. Supplemental S1A&B: One would like to see quantitation of nephron formation.
4. Fig3: is there some tissue on the outside of Aox3 expression? Is it not on the outermost periphery?
5. Line 272: Typo, should be Figure 3

Reviewer 2*Advance summary and potential significance to field*

The nephrogenic zone of the developing kidney includes 3 co-dependent progenitor populations; Nephron progenitors, stromal progenitors and collecting duct tip. Many studies have demonstrated that the co-dependence of these cell populations depends on signaling between them, but few studies have addressed communication between nephron progenitor cells and stromal progenitors. In this work, the authors use the nephron progenitor cell restricted Six2cre to disrupt the nephron progenitor lineage at distinct stages of differentiation; inducible DTA largely removes cells of the lineage, Wnt4 conditional maintains nephron progenitor cells but prevents their differentiation, and Wt1 conditional perturbs both nephron progenitor cells and their differentiation. They find an expansion of the stromal progenitor population in all 3 of these strains, indicating that the nephron lineage restricts stromal progenitors. They then investigate snRNA-seq of 3 Wilms tumor samples and find similarities with human fetal kidney stroma.

The objective of the study is to ask if the nephron lineage controls the stromal cell differentiation program, so the null hypothesis would be that stromal differentiation is independent of the nephron lineage. Considering the size reduction in mutant kidneys, lack of any evidence for increased proliferation in the stromal compartment, and largely preserved gene expression profile of stromal cells, it seems plausible that the stromal progenitor number is correct for the developmental stage analyzed but because mutant kidneys are smaller, stroma must be distributed in a smaller volume and thus it forms layers. So, in my opinion the null hypothesis remains equally likely considering the data presented, and the authors need to provide stronger proof for nephron lineage control of stromal differentiation if they want to draw that conclusion. Regarding the Wilms tumor data, too little information is provided for the reader to interpret it.

Comments

According to the text, all mutant kidneys are smaller, but no quantitative data is provided. One possibility is that stromal progenitors reach their correct number for the stage of development but are distributed in a smaller volume resulting in the layering that is seen. Proliferation marker analyses in S1 do not show any obvious difference between the peripheral stroma in mutant and wild type which would support this interpretation. But quantitative data on sizes of the kidneys analyzed in the study as well as a quantitative evaluation of proliferation index is needed to interpret the findings.

The collecting duct is never analyzed, although the strong expression of Gdnf and Aldh1a2 suggest perturbation in that compartment which may feed back on stromal progenitors. The proliferation markers shown in S1 would support that - epithelial tubules resembling collecting ducts have elevated PCNA and pHH3. Proliferation index and collecting duct marker analysis is necessary to understand the behavior of the collecting duct in these mutants.

Given the data presented, I do not agree that subcluster 7 of the stroma is abnormally expanded. In figure 3, only genes expressed in subcluster 7 are analyzed, so there is no basis to say it this particular subcluster is abnormally expanded. You would only be able to say this if subcluster 7 was disproportionately expanded relative to other subclusters in the mutant but there is no data to answer that. This may just be showing appropriate stromal lineage differentiation but smaller organ volume resulting in increased stromal cell numbers of all subclusters relative to the nephron lineage.

Conclusions from thin sections about abundance of stroma in NPC-depleted niches are not really convincing since the NPCs may simply be in a different plane. This would need to show in whole-mount/3D or in series of serial sections so you could compare niches with and without NPCs.

The point being made regarding the Wilms tumor analysis is obscure because there is a lack of comparators and information. At what age were these resections performed & what profile of stromal cells would be expected in healthy tissue? The finding that stromal cells in Wilms tumor show transcriptional similarities to human fetal kidney stroma would only really be significant if it was known that age-matched healthy kidney stroma did not show similarities with fetal stroma. The paper that the authors refer to as the source for the tissue reports a large number of tumors and normal tissues so they need to be specific about the subset of the samples used in this manuscript.

Ectopic expression of NCAM is the most striking observation in the study regarding stromal cells, but it is not explored. It should at least be discussed.

Make sure the figure panels are cited in order.

Review the text to make sure the figure citations are correct throughout.

Limitations of interpreting data from the Six2cre transgene are not acknowledged but should be. This paper should be discussed:

Perl AJ et al. Reduced Nephron Endowment in Six2-TGCTg Mice Is Due to Six3 Misexpression by Aberrant Enhancer-Promoter Interactions in the Transgene. *J Am Soc Nephrol*. 2024 May 1;35(5):566-577.

These papers have reported similar observations to the data in this manuscript and should be discussed:

Cebrian C et al. The number of fetal nephron progenitor cells limits ureteric branching and adult nephron endowment. *Cell Rep*. 2014 Apr 10;7(1):127-37.

Muthukrishnan SD et al. Nephron progenitor cell death elicits a limited compensatory response associated with interstitial expansion in the neonatal kidney. *Dis Model Mech*. 2018 Jan 29;11(1).

Reviewer 3

Advance summary and potential significance to field

Michalopoulos et al demonstrate that disruption of the nephron progenitor population via Wt1 ablation results in expansion of the stromal progenitor population which is phenocopied by Wnt4-null mutants and nephron progenitor ablation via diphtheria toxin. A subpopulation of FoxD1-positive progenitor cells also exists in human fetal kidney and human Wilms tumor samples by single nucleus RNAseq. The paper is well-written, the experiments are well-designed, the images for immunostaining and ISH are excellent, and the data are robust. The paper describes the renal stromal phenotype associated with three mouse models in which there is an expansion of the renal stroma, snRNAseq showing distinct subclusters of FoxD1-positive cells and validation via ISH, re-analysis of human fetal kidney snRNAseq data for renal stromal populations and performed snRNAseq on 3 Wilms tumors samples. The paper describes markers for a subcluster of presumed renal stromal progenitors.

My major concern with this paper is that the observations around the three transgenic mouse models are incremental to the field and the snRNAseq as presented is largely descriptive. As the authors cite, a conditional Nestin-cre, Wt1 floxed mouse has been reported where Wt1 is deleted from the metanephric mesenchyme, resulting in an expansion of the stroma (as seen in the Six2-Cre, floxed Wt1 kidneys). Though this paper provides additional information about the renal stroma in Wnt4 null mice and in the setting of ablation of nephron progenitors via diphtheria toxin, the Wnt4 null mouse has been reported previously, as has a Gdnf-cre, DTA mouse (albeit without the focus on kidney stroma). Overall, the observation that there is an expansion of renal stroma in the setting of loss of nephron progenitors has also been observed in other mouse models in which there is a loss of nephron progenitors (as cited by the authors, in Six2, mdm mice, amongst others).

Though markers of presumed renal stromal progenitors are described, there is little to support a mechanism underlying stromal-progenitor crosstalk.

Other comments:

1. Cre-negative mice were used as controls. The Six2-TGC allele used in this study is known to have a renal hypoplasia phenotype, so difficult to exclude whether some of the observed phenotypes are related to the Six2-Cre allele. In particular, the snRNAseq experiment in Figure 2 does not account for this as the samples are pooled. Is there a difference between the Cre-negative and Six2-TGC kidneys, although the sample numbers are relatively small?
2. The observation of expansion of the renal stroma is primarily on the basis of morphology - given that many of the identified markers are stromal-specific, would be helpful to both validate the snRNAseq data and to quantify the expansion with other quantitative measures, eg. qPCR or Western.

First revision

Author response to reviewers' comments

Dear Editors and Reviewers,

We would like to thank the reviewers for their thoughtful comments and critiques and for the opportunity to provide this resubmission. As outlined in the point-by-point detailed response below, we have significantly revised the manuscript with additional data/experiments, including: 1) quantification of kidney size and additional data evaluating the *Six2cre;RosaDTA^{cl+}* model to provide further insights into potential mechanisms driving the expansion of the stromal progenitor population, and 2) single nuclei RNA-seq of the *Six2cre;Wt1^{cl/c}* model, as this demonstrates transcriptional changes in the *Foxd1+* stromal progenitor subclusters compared to *Six2cre* control kidneys, with a number of identified genes validated by in-situ hybridization across the three mutant models. Overall, the addition of these experiments provides a more comprehensive analysis of the nephrogenic stroma in mutants kidneys with defects in nephrogenesis, as requested by the reviewers in the initial critique. Given the inclusion of this new data and the concerns raised in the initial review regarding the limitations of the Wilms tumor data, the tumor stroma analysis has subsequently been removed. In summary, we very much appreciate the opportunity to provide this revision, which we hope you agree has resulted in a stronger, more focused manuscript as detailed in the point-by-point response below and shown in the “highlighted” version of the revised manuscript with major changes to the text/figures denoted in blue text.

Reviewer 1:

This paper aims to understand how signals from the developing nephrogenic lineage regulate stromal development. To address this question, three mutant mouse lines were generated: *Six2Cre-Wt1* cKO, loss of *Wnt4* which is a downstream effector of *Wt1* in regulation of nephrogenesis, and NP ablation by *Six2Cre-DTA*, each introducing disruptions in nephrogenic lineage development.

Wt1 is known to maintain the metanephric mesenchyme and drive the nephrogenic process in the embryonic kidney. Loss of *Wt1* results in widespread apoptosis in developing kidney and eventually kidney agenesis. Mutated *WT1* is related to formation of Wilm's Tumor. In this study, all three mutant mouse lines exhibited anomalies in nephron progenitors and nephrogenic intermediates, to a variable extent, and demonstrated an expanded cell population located at the outer periphery of kidneys in the place of stromal progenitors. Age-matched control kidneys were then analyzed by single-nuclei RNA sequencing, which identified subpopulations of stromal progenitors. Analyses in mutant kidneys then confirmed that the expanded peripheral population expresses some identified markers of stromal progenitors, and that this expanded population is not due to loss of nephron progenitor cells. The authors then analyzed three Wilm's Tumor samples by single-nuclei RNA sequencing to compare with control human fetal kidneys and control mouse kidneys, and discovered that the majority of stromal cells from Wilm's Tumor samples are transcriptionally similar to stromal progenitors.

SUGGESTIONS TO AUTHORS

Major comments:

1. In the absence of a more comprehensive analysis of stromal cells in mutant kidneys and their gene expression, one does not know what the expanded population represents. That is, does it represent an increased number of normal stromal cells, or does it represent an unusual population of stromal cells? While some stromal progenitor markers are expressed in the expanded population, it is not very likely that this population simply results from increased normal stromal cells, since (1) NCAM is ectopically expressed in this population, and (2) cell proliferation is unchanged in the stroma of mutant kidneys.

We very much appreciate this critique. In response, we performed single nuclei RNA-seq of the *Six2cre;Wt1^{cl/c}* mutant mouse model and have provided new data comparing mutants to control *Six2cre* kidneys as shown in Figure 2 and Figure 5. As supported by this data, we now clarify that the *Wt1* model shows 1) an expansion of nephrogenic stromal cells that maintain expression of a significant number of “anchor/marker” genes, and 2) abnormal gene expression changes in the *Foxd1+* stromal cells of *Six2cre;Wt1^{cl/c}* mutant kidneys, likely related to the lack of signaling from defects in nephrogenesis. For example, mutant kidneys show increased *Ncam2* - likely accounting for the IF showing ectopic NCAM staining in the *Wt1* mutant model (as this antibody recognized both NCAM1 and NCAM2) as well as other differentially expressed genes across the three *Foxd1+* subclusters, as shown in Figure 5. Taken together, these findings suggest both an increased cell number/expansion as well as molecular dysregulation of the *Foxd1+* stromal population due to a block nephrogenesis in the *Six2cre;Wt1^{cl/c}* model, and this has been further clarified in both the results (lines 222-224, 249-259, and 341-361) and the discussion (lines 409-415 and 440-447).

We acknowledge that we only provide single nuclei RNA-seq from one mutant model (i.e., *Six2cre;Wt1^{cl/c}* mutant), given the time/cost limitations of generating and analyzing single nuclei RNA-seq from the two additional models included in this report. However, our extensive validation studies via ISH and IF in both the *Wnt4-null* and *Six2cre;RosaDTA* models of a number of genes localized to the *Foxd1+* subclusters identified in the control and *Six2cre;Wt1^{cl/c}* model nonetheless provide important support the overall finding that the nephrogenic stroma becomes expanded when nephrogenesis is disrupted, thus confirming that this finding is not unique to the *Six2cre;Wt1^{cl/c}* mutant model. Furthermore, the “deeper dive” now provided in the revision evaluating the molecular changes of the *Foxd1* stromal progenitor population specifically in the *Six2cre;Wt1^{cl/c}* mutant model is of potential clinical interest, given the developmental pathologies specifically associated with human WT1 mutations, thus the rationale for providing the data specifically for this model.

2. While this study demonstrates that an abnormal nephrogenic lineage can affect the stromal lineage, it did not reveal the mechanisms underlying this interaction. What is the nature of this non-cell-autonomous regulation? How do abnormalities in the nephrogenic lineage affect stromal development?

The reviewer is correct that a limitation of this study is that we have yet to identify the underlying molecular mechanism(s) coordinating normal stromal development during nephrogenesis. Nonetheless, our findings provide scientific rigor demonstrating that the stromal progenitor population undergoes “abnormal” developmental changes in the absence of normally developing nephrons (as further supported by the single nuclei RNA-seq data described above). Further, this report provides insight into the cellular mechanisms directing the abnormal expansion through the examination of three distinct genetic models and additional data on the *Six2cre;RosaDTA^{cl/c}* mutant model additionally provided in this resubmission. Specifically, we show that *Foxd1+* stromal progenitor population undergo a “multilayering expansion” in the absence of developing nephrons (in the *Six2cre;Wt1^{cl/c}* and *Wnt4-null* models) as well as NPCs/UB (in the *Six2cre;RosaDTA* model). Additionally, we show the kidney size of the mutant models vs controls and extensively characterized the *Six2cre;RosaDTA* model changes throughout development. Now included in the discussion (lines 435-449), this data suggests that the *Foxd1+* stromal progenitor population “cell-autonomously” expands during early stages of development (i.e., up to E15.5 as suggested by our analysis) irrespective of signals from the NPCs/UB but then subsequently “regresses” at later stages of development (i.e., E18.5) as shown in Figure 4.

Furthermore, single nuclei RNA-seq data from mutant *Six2cre;Wt1^{cl/c}* and control *Six2cre* kidneys in this revised manuscript shows abnormal gene expression in the mutant *Foxd1+* progenitor population. While we have provided additional discussion regarding potential mechanisms of this non-cell autonomous regulation, we also acknowledge that this remains a limitation of our study (lines 440-447 and 470-474) warranting additional future investigations.

3. To address these questions above, one would like to see the transcriptome of the mutant kidneys being analyzed at a single-cell level, with special focus on stromal cells, to compare with the presented control data. This experiment is necessary given that these questions are at the heart of the study.

As per the reviewer's initial comment (i.e., comment 1 described above), we have performed single nuclei RNA-seq of the *Six2cre;Wt1^{cl/c}* mutant mouse model, as this mutant has clinical relevance to developmental diseases that may arise from WT1 mutations in humans. Although performing single nuclei RNA-seq of all three models included in this report is beyond the scope of this current manuscript, we feel that inclusion of this data from *Six2cre;Wt1^{cl/c}* mutant mouse significantly strengthens our previous data demonstrating abnormal cellular and molecular changes in the *Foxd1+* stromal progenitor population, as detailed in response to the initial comment above.

4. While the comparison between Wilm's Tumor samples and healthy control kidneys is interesting, this part alone is not very helpful in understanding the abnormal interaction between nephrogenic cells and stromal cells demonstrated by mutant kidneys, as the connection between tumor samples and mutant mouse kidneys remains questionable until the anomalies in the mutant mouse kidneys have been fully characterized.

We very much appreciate this comment and agree with the reviewer that there are limitations in our analysis of the human Wilms tumor data, so this data has been removed from the manuscript.

5. It is understood that the tumor samples have some similarities to mutant kidneys in terms of expanded stroma. However, only one patient sample was identified as positive for WT1 mutation, and the one that exhibits nephrogenic rests is negative for a WT1 mutation. Moreover, unlike patients that carry global mutations, mutant mice carry lineage-specific mutations in the nephrogenic lineage. How does this WT1 variant affect the function of WT1 at molecular level? Does it lead to a complete loss of function similar to Cre deletion? To what degree do the mutant mouse kidneys and human Wilm's Tumor mimic each other?

As the reviewer points out, we only have access to one Wilms tumor sample with a known WT1 mutation, and since comparisons of our mutant mouse models to Wilms tumor samples is outside of the scope/focus of this manuscript, we have removed the tumor data. Regarding the *Wt1* mutation used in this model, we clarify that exon 8-9 is deleted, thus targeting the DNA binding domain and expected transcriptional activity (lines 122-123).

6. Line 151-153: One would ask: are these cells really derived from stromal progenitors? Couldn't it be maldeveloped nephrogenic cells that gained some stromal gene expression moving outwards, given that they are NCAM+? The YFP lineage tracing (line 146-148) is very helpful in addressing this issue but this needs to be explicitly discussed.

We have clarified that inclusion of the YFP reporter specifically targets the nephron lineage, with no recombination in the stroma and also arguing against NPC "transdifferentiating" into stromal cells (lines 144-145).

7. Line 362-363: data integration can mask ectopic population as this step forcefully maps populations between datasets. It is understood that correction is inevitable before combining datasets but this limitation needs to be discussed.

We appreciate the reviewer identifying this limitation in the analysis of single nuclei RNA-seq data. While we have removed the Wilms tumor data from this manuscript, the same issue may apply to our analysis of the stroma from control and *Six2cre;Wt1^{cl/c}* mutant kidneys. One way to try to address this is to perform stringent unsupervised clustering to try to pick up any differences among

the captured cell types/clusters. When we analyzed our data at this “highest resolution” (i.e., 1.5 vs 0.9 - which is presented in the paper), we identify four subclusters of *Foxd1*+ stroma and all four still include cells from the control sample, again further arguing against the formation of a “distinct” cell type. Also, in the supplemental material, we provide extensive details on all the clusters identified in the dataset, including those excluded from the final analysis (as shown in Supplemental Fig. S2) to ensure we did not exclude a cluster specific to the mutant dataset.

Minor comments:

1. One would like to see confirmation of loss of *Wt1*.

We appreciated this suggestion; however, this is technically challenging, given that a mutant protein is still made in our model (as we are only targeting exon 8-9) and this protein is recognized by the commercially available *WT1* antibody. However, we consistently observe a mutant phenotype (including the transcriptomic alterations described) that closely resembles the previously published studies of *Wt1* deletion in NPCs and our transcriptomic data.

2. Line 130-131: please elaborate the anomalies in nephrogenic structures. While *LHX1*+ structures are absent, it seems that *NCAM1* is present in all three mutants.

We have clarified that a few differentiating structures (that are *NCAM* positive) do form in the mutant mouse models; however, the self-renewing NPCs in *Six2cre;Wt1^{cl/c}* mutant kidneys actually do not show *NCAM* expression (shown both in Fig. 1, N and Supplemental Fig. S1, H).

3. Supplemental S1A&B: One would like to see quantitation of nephron formation.

While we have not provided nephron quantification in our mutant mouse models (given that these mice do not survive past postnatal day 1), we have now included single nuclei RNA-seq data at E15.5 that shows decreased capture of podocytes, further supporting a defect in nephrogenesis.

4. Fig3: is there some tissue on the outside of *Aox3* expression? Is it not on the outermost periphery?

The outer edge of the kidney is shown in each panel - while *Aox3* expression is somewhat weaker than the other *ISH* probes, the images are representative of the nephrogenic zone stroma.

5. Line 272: Typo, should be Figure 3

We thank the reviewer for identifying this typo - it has been corrected.

Reviewer 2:

The nephrogenic zone of the developing kidney includes 3 co-dependent progenitor populations; Nephron progenitors, stromal progenitors and collecting duct tip. Many studies have demonstrated that the co-dependence of these cell populations depends on signaling between them, but few studies have addressed communication between nephron progenitor cells and stromal progenitors. In this work, the authors use the nephron progenitor cell restricted *Six2cre* to disrupt the nephron progenitor lineage at distinct stages of differentiation; inducible *DTA* largely removes cells of the lineage, *Wnt4* conditional maintains nephron progenitor cells but prevents their differentiation, and *Wt1* conditional perturbs both nephron progenitor cells and their differentiation. They find an expansion of the stromal progenitor population in all 3 of these strains, indicating that the nephron lineage restricts stromal progenitors. They then investigate snRNA-seq of 3 Wilms tumor samples and find similarities with human fetal kidney stroma.

The objective of the study is to ask if the nephron lineage controls the stromal cell differentiation program, so the null hypothesis would be that stromal differentiation is independent of the nephron lineage. Considering the size reduction in mutant kidneys, lack of any evidence for increased proliferation in the stromal compartment, and largely preserved gene expression profile of stromal cells, it seems plausible that the stromal progenitor number is correct for the developmental stage analyzed but because mutant kidneys are smaller, stroma must be distributed in a smaller volume

and thus it forms layers. So, in my opinion the null hypothesis remains equally likely considering the data presented, and the authors need to provide stronger proof for nephron lineage control of stromal differentiation if they want to draw that conclusion. Regarding the Wilms tumor data, too little information is provided for the reader to interpret it.

Comments:

According to the text, all mutant kidneys are smaller, but no quantitative data is provided. One possibility is that stromal progenitors reach their correct number for the stage of development but are distributed in a smaller volume resulting in the layering that is seen. Proliferation marker analyses in S1 do not show any obvious difference between the peripheral stroma in mutant and wild type which would support this interpretation. But quantitative data on sizes of the kidneys analyzed in the study as well as a quantitative evaluation of proliferation index is needed to interpret the findings.

We very much appreciate this critique and have subsequently provided both qualitative data (Fig. 1, A-D showing whole kidney images) and well as quantification of kidney size (Fig. 4, E and Supplemental Fig. S1, M showing measured kidney length). Additionally, we provide quantification of the cycling/proliferating cells via single-nuclei RNA-seq including three replicates sequencing runs with > 10,000 nuclei for each genotype, providing a robust dataset to make inferences on the proportions of cell types captured. With this data, we now have clarified in the manuscript that stromal progenitor proliferation may occur as a cell-autonomous mechanism (regardless of overall kidney size or signals from the nephrogenic niche) which may result in an “accumulation” of the stromal progenitor population unable to differentiate due to a lack of signals from normally developing nephrons. However, the lack of continued or increasing expansion of stromal progenitor cells in later developmental timepoints (i.e., E18.5) in *Six2cre;RosaDTA^{cl/+}* mutant kidneys suggests dynamic, time-dependent changes in the *Foxd1+* population. These additional findings have been added to the manuscript in the results section (lines 302-304 and 329-336) and discussion (434-447).

The collecting duct is never analyzed, although the strong expression of *Gdnf* and *Aldh1a2* suggest perturbation in that compartment which may feed back on stromal progenitors. The proliferation markers shown in S1 would support that - epithelial tubules resembling collecting ducts have elevated PCNA and pHH3. Proliferation index and collecting duct marker analysis is necessary to understand the behavior of the collecting duct in these mutants.

We also very much appreciate this critique. Given that the expansion of *Foxd1+* stromal progenitor cells occurs in *Six2cre;RosaDTA^{cl/+}* mutant kidneys even in regions lacking NPCs/UB, this would suggest that the expansion does not rely on crosstalk from other progenitor populations in the nephrogenic niche. However, we do highlight this possibility as a mechanism in the *Six2creWt1^{cl/c}* and *Wnt4-null* mutant models in the discussion (lines 455-460) and plan to evaluate this in subsequent studies.

Given the data presented, I do not agree that subcluster 7 of the stroma is abnormally expanded. In figure 3, only genes expressed in subcluster 7 are analyzed, so there is no basis to say it this particular subcluster is abnormally expanded. You would only be able to say this if subcluster 7 was disproportionately expanded relative to other subclusters in the mutant but there is no data to answer that. This may just be showing appropriate stromal lineage differentiation but smaller organ volume resulting in increased stromal cell numbers of all subclusters relative to the nephron lineage.

To further demonstrate changes in the *Foxd1+* stromal subclusters, we now include single nuclei RNA-seq of the *Six2cre;Wt1^{cl/c}* mutant mouse model in comparison to control *Six2cre* kidneys as shown in Figure 2 and Figure 5 (as described above in response to the critique from reviewer 1). This robust dataset including triplicates of each genotype provides additional data on the proportions of cell types captured (as shown in Supplemental Fig. S3 and Supplemental Table S1). This data, along with the ISH validation, suggests that all subclusters of the *Foxd1+* progenitor population are increased. We also clarify how continued proliferation of the stromal progenitors at earlier time points in development in the setting of reduced organ size (as the reviewer points out here) may contribute to the abnormal *Foxd1+* cellular expansion, as described above in the response to the reviewer’s initial comment.

Conclusions from thin sections about abundance of stroma in NPC-depleted niches are not really convincing since the NPCs may simply be in a different plane. This would need to show in whole-mount/3D or in series of serial sections so you could compare niches with and without NPCs.

We have evaluated serial sections to ensure that regions devoid of NPCs/UB are not artifacts of the single, thin section and have confirmed stromal expansion in these regions (lines 313-315).

The point being made regarding the Wilms tumor analysis is obscure because there is a lack of comparators and information. At what age were these resections performed & what profile of stromal cells would be expected in healthy tissue? The finding that stromal cells in Wilms tumor show transcriptional similarities to human fetal kidney stroma would only really be significant if it was known that age-matched healthy kidney stroma did not show similarities with fetal stroma. The paper that the authors refer to as the source for the tissue reports a large number of tumors and normal tissues so they need to be specific about the subset of the samples used in this manuscript.

We agree with the reviewer that there are limitations in our analysis of the human Wilms tumor data, so this data has been removed from the manuscript.

Ectopic expression of NCAM is the most striking observation in the study regarding stromal cells, but it is not explored. It should at least be discussed.

To address this important point raised by the reviewer, we evaluated the single nuclei RNA-seq data of control and *Six2cre;Wt1^{cl/c}* mutant kidneys and found no changes in *Ncam1* expression but did see significantly increased expression of *Ncam2* (log2FC 1.42, adj p-value = 8.5E-82) as shown in Fig 5 panels F and G. The antibody utilized for IF is reported to recognize both NCAM1 and NCAM2 (<https://www.sigmaaldrich.com/US/en/product/sigma/c9672>), likely accounting of the increased staining observed by IF. This has been updated in the manuscript results (lines 352-354) and methods (lines 523-524).

Make sure the figure panels are cited in order.

This has been corrected, and figures have been updated to reflect the additional data.

Review the text to make sure the figure citations are correct throughout.

This has also been corrected.

Limitations of interpreting data from the *Six2cre* transgene are not acknowledged but should be. This paper should be discussed:
Perl AJ et al. Reduced Nephron Endowment in *Six2*-TGCTg Mice Is Due to *Six3* Misexpression by Aberrant Enhancer-Promoter Interactions in the Transgene. *J Am Soc Nephrol*. 2024 May 1;35(5):566-577.

We appreciate this suggest and have corrected this oversight by including the reference and rationale for the use of *Six2cre* mice as controls (lines 222-224).

These papers have reported similar observations to the data in this manuscript and should be discussed:
Cebrian C et al. The number of fetal nephron progenitor cells limits ureteric branching and adult nephron endowment. *Cell Rep*. 2014 Apr 10;7(1):127-37.
Muthukrishnan SD et al. Nephron progenitor cell death elicits a limited compensatory response associated with interstitial expansion in the neonatal kidney. *Dis Model Mech*. 2018 Jan 29;11(1).

We very much appreciate these suggestions. The paper by Cebrian et al used a tamoxifen-inducible *Gdnf-CreERT2* mouse line to generate a 40% reduction in the number of fetal nephron progenitor cells and found that these kidneys “self-corrected” by decreasing UB branching to allow NPC numbers to recover, but the UB branching defect persisted and resulted in reduced nephron endowment. Effects on the stroma were not analyzed in this report (and may be confounded given

the stromal expression of Gdnf), so this citation has been added to the introduction referencing NPC-to-UB crosstalk (line 74).

The paper by Muthukrishnan et al similarly depleted approximately 40% of nephron progenitor cells during fetal development and reported a 10-15% reduction in nephrons. They additionally reported an expansion of renal interstitium localized to the outer medullary region with increased proliferation of these cells at postnatal time points; however, after analyzing a second model, these findings were thought to be a unique response to apoptosis-induced cellular depletion and not caused by loss of NPCs or the pathophysiological changes associated with reduced nephron number. Given that this report did not specifically evaluate for changes in the stromal progenitor cells with NPC depletion, the citation has been added to the discussion on how the stromal cells function in lineage crosstalk in the kidney during development and response to injury (line 483).

Reviewer 3:

Michalopoulos et al demonstrate that disruption of the nephron progenitor population via Wt1 ablation results in expansion of the stromal progenitor population which is phenocopied by Wnt4-null mutants and nephron progenitor ablation via diphtheria toxin. A subpopulation of FoxD1-positive progenitor cells also exists in human fetal kidney and human Wilms tumor samples by single nucleus RNAseq. The paper is well-written, the experiments are well-designed, the images for immunostaining and ISH are excellent, and the data are robust. The paper describes the renal stromal phenotype associated with three mouse models in which there is an expansion of the renal stroma, snRNAseq showing distinct subclusters of FoxD1-positive cells and validation via ISH, re-analysis of human fetal kidney snRNAseq data for renal stromal populations and performed snRNAseq on 3 Wilms tumors samples. The paper describes markers for a subcluster of presumed renal stromal progenitors.

My major concern with this paper is that the observations around the three transgenic mouse models are incremental to the field and the snRNAseq as presented is largely descriptive. As the authors cite, a conditional Nestin-cre, Wt1 floxed mouse has been reported where Wt1 is deleted from the metanephric mesenchyme, resulting in an expansion of the stroma (as seen in the Six2-Cre, floxed Wt1 kidneys). Though this paper provides additional information about the renal stroma in Wnt4 null mice and in the setting of ablation of nephron progenitors via diphtheria toxin, the Wnt4 null mouse has been reported previously, as has a Gdnf-cre, DTA mouse (albeit without the focus on kidney stroma). Overall, the observation that there is an expansion of renal stroma in the setting of loss of nephron progenitors has also been observed in other mouse models in which there is a loss of nephron progenitors (as cited by the authors, in Six2, mdm mice, amongst others). Though markers of presumed renal stromal progenitors are described, there is little to support a mechanism underlying stromal-progenitor crosstalk.

We appreciate the reviewers overall positive comments and would like to highlight that the previous publication of the Nestin-cre model examining targeted loss-of-function of Wt1 mutations did not report any changes in the stroma, and it is unclear if the authors did not observe this phenotype or did not evaluate for it. Additionally, while the paper characterizing the Mdm2 knock out model does report an expansion of the nephrogenic stroma, this is limited to one figure and did not investigate the observation further. To our knowledge, this Mdm2 KO model is the only report (prior to our work here) that shows expansion of the stromal progenitor cells in mouse models with defects in nephrogenesis. Thus, our study is the first to detail the nature of the expansion including multiple models to evaluate abnormalities in mutant NPCs vs the loss of NPCs vs the loss of differentiating nephron structures and therefore feel it would be of interest to the kidney development field.

Other comments:

3. Cre-negative mice were used as controls. The Six2-TGC allele used in this study is known to have a renal hypoplasia phenotype, so difficult to exclude whether some of the observed phenotypes are related to the Six2-Cre allele. In particular, the snRNAseq experiment in Figure 2 does not account for this as the samples are pooled. Is there a difference between the Cre-negative and Six2-TGC kidneys, although the sample numbers are relatively small?

We appreciate this comment from the reviewer. We did not detect significant gene expression differences in the stroma of cre-negative vs Six2cre-TGC only single nuclei RNA-seq, so we subsequently used Six2cre mice as controls for these experiments.

4. The observation of expansion of the renal stroma is primarily on the basis of morphology - given that many of the identified markers are stromal-specific, would be helpful to both validate the snRNAseq data and to quantify the expansion with other quantitative measures, eg. qPCR or Western.

To further evaluate the cellular expansion and transcriptional changes in the stroma, we now include single nuclei RNA-seq of control and Six2cre;Wt1^{c/c} mutant kidneys, as detailed above in the response to the critiques from reviewers 1 and 2, which we feel provides a more comprehensive analysis of the changes in stromal development that occur due to defects in nephrogenesis.

Second decision letter

MS ID#: dev.204964R1

MS TITLE: Defects in nephrogenesis result in an expansion of the Foxd1+ stromal progenitor population

AUTHORS: Michael G. Michalopoulos, Yan Liu, Dinesh Ravindra Raju, John T. Labin, Yanru Ma, Dhruv Gaur, Sadiksha Khadka, Chao Xing, Andrew P. McMahon, Thomas J. Carroll and Keri Anne Drake

Dear Dr Drake,

I am happy to tell you that your manuscript has been accepted for publication in Development, pending our standard publication integrity checks.

Reviewer 1

Advance summary and potential significance to field

Summary:

The authors have revised the original manuscript with the addition of a single-nuclear RNA sequencing analysis of the Wt1 cKO model and data from other models, including quantitation of kidney size and histological data from the Six2-DTA model. Data from human kidneys and tumor samples have been removed. While this revision strengthens the previous observation that the stromal progenitor population is indeed expanded, all additional data remain descriptive. Hence the revised manuscript has not addressed the main question/concern raised upon review of the original manuscript: what are the molecular and cellular mechanisms underlying the abnormal expansion of the stromal progenitors?

Follow up on comments arising from review of the original manuscript (#s refer to numbered comments related to the original manuscript):

Major comments

#1: The additional sn RNA seq analysis does not provide additional information on what exactly is happening in the expanded stromal population; the authors mainly focused on the hallmark genes that are used to establish the clusters. This approach is unacceptable as it is biased towards the hypothesis that the anomaly is only due to disruption in differentiation/specification of stromal subpopulations, which is not necessarily the case. The authors only listed a subset of differentially expressed genes in stromal clusters without discussing what they mean. Hence the sequencing data provided is descriptive rather than mechanistic and does not help establish the underlying cellular and molecular mechanisms. DEGs need to be analyzed in each cluster, with special attention to those in which anomalies have been observed (e.g., expanded stromal cluster 14, 1, 7, and 19, and

decreased nephrogenic cluster 13, 3, 16, and 8). In each cluster, DEGs need to be analyzed for pathway and gene ontology enrichment. Identified targets need to be validated by functional tests in developing kidneys. Without this workflow, the authors have potentially overlooked many potential targets that are worth exploring (e.g., DEGs from combined cluster 14, 1, and 7, provided in Supplemental Table S4, demonstrate upregulation in cell-cell adhesion and downregulation in mitosis),

Of the three models generated to target the nephrogenic lineage at different developmental stages, the authors only investigated the Wt1 cKO model with snRNA seq. The snRNA seq data combined with histological data clearly demonstrate a disruption in nephrogenic intermediates in the Wt1 cKO model. Therefore, it is unclear if the observed stromal anomalies result from disrupted signals from the nephron progenitors or malformed nephrogenic intermediates. Without a more detailed analysis on the Wnt4 null model, the provided data is insufficient to establish the cellular and molecular mechanisms underlying the nephrogenic-stromal cross-talk.

#2: The provided data does not show the cellular mechanisms underlying the expanded stroma as no experiment was performed to characterize alterations in cellular behavior. The term "multilayering" is confusing and has not been clearly defined.

The authors described the stromal expansion as "cell-autonomous". For the following reasons, the provided data is insufficient to support this conclusion that is very unlikely to be true.

1. The term "cell-autonomous" was used to describe expansion of stromal progenitor cells in the Six2-DTA model and implied that the stromal progenitors are unaffected but distributed in a smaller kidney. Thus, the mutant kidneys demonstrate a relatively expanded stroma. The authors reasoned that stromal progenitors are present in areas where nephron progenitors are fully ablated. However, presence of stromal progenitors is not equivalent to normal expansion. Since excessive expansion is not distinguished from normal expansion, maintenance, and self-renewal, it is problematic to draw such a conclusion, as one does not know if expansion of stromal progenitors is truly normal or not.

2. Although the expansion of stromal progenitors in the Six2-DTA model was not characterized, sn RNA seq data from the Wt1 cKO model demonstrates not only an increased percentage of stromal progenitors relative to whole kidney and all stromal cells, but also an increase in the absolute number of stromal progenitors, with no decrease in the number of stromal derivatives. Therefore, expansion of stromal progenitors in the Wt1 cKO model is clearly abnormal. Given that the three mouse models have similar phenotypes, it is unlikely that stromal progenitor expansion in the Six2-DTA model is normal and cell-autonomous.

3. While stromal progenitors are maintained in kidneys with nephrogenic defects, it seems that the stromal progenitors are inhibited from excessive expansion by inhibitory signals from the nephrogenic lineage. To conclude that stromal expansion is truly independent, in the sn RNA seq data the nephrogenic lineage needs to be carefully examined, and cell communication analysis needs to be performed to show that there is no alteration in signals from the nephrogenic lineage in mutant kidneys, which is very unlikely.

#3: see #1 above.

Minor comments

#5: this has not been corrected.

Additional comments regarding the latest data provided in the revision.

Major comments:

1. The DEGs in combined stromal progenitors (combined cluster 14, 1, and 7) as listed in Supplemental Table S4 ($\log_2FC > 0.58$, adjusted P value < 0.05) show decreased mitosis. This is consistent with the fact that within stromal progenitors (combined cluster 14, 1, and 7), the proportion of cycling cells (cluster 7) is decreased, but contradicts the observed stromal expansion. More experiments are needed to reconcile these conflicting findings, and a related discussion is required.

2. It seems that maintenance of the expanded stroma is related to ureteric signals (Fig 4I). It is unclear whether expansion of stromal progenitors is directly regulated by nephrogenic defects or a secondary effect mediated by abnormal signals from the ureteric lineage. Therefore, a detailed analysis of the ureteric populations in the sn RNA seq data is necessary.

3. The authors screened DEGs with a $\log_2FC > 0.2$. This is equivalent to a fold-change of 1.14. This is not a commonly used cut-off. Normally, DEGs are selected with FC of 1.5 ($\log_2FC = 0.58$) or 2 ($\log_2FC=1$). The reason for choosing $\log_2FC > 0.2$ needs to be explained.

4. One gets a vague impression that cluster 14 is special, but it is neither the only expanded population nor the only population demonstrating transcriptional alterations. While it is conserved in human and mouse NORMAL kidneys, there doesn't seem to be data demonstrating how this cluster is special in relation to stromal progenitor expansion in the context of nephrogenic disruption.

Minor:

1. Please make sure figures and panels are referenced correctly.
2. The term "nephrogenic stroma" is confusing, as one may understand it as stromal progenitor cells that give rise to nephrons. Are the authors trying to refer to stromal progenitor cells?

Reviewer 2

Advance summary and potential significance to field

This paper shows that the stromal population surrounding nephron progenitor cells expands when the nephron lineage is handicapped using 3 different models. The central observation is based on immunostaining, and the authors provide supportive data using single cell transcriptomics. Their findings imply that there is an axis of communication between the nephron lineage and the surrounding stroma, which is significant.

Comments for the author

My questions have been answered in this revision.